# On the detrimental effect of invariances in the likelihood for variational inference

**Richard Kurle** [*]
AWS AI Labs
`kurler@amazon.com`

**Ralf Herbrich**
Hasso-Plattner Institut
`ralf.herbrich@hpi.de`

**Tim Januschowski** [†]
Zalando SE
`tim.januschowski@zalando.de`

**Yuyang Wang**
AWS AI Labs
`yuyawang@amazon.com`

**Jan Gasthaus**
AWS AI Labs
`gasthaus@amazon.com`

## Abstract

Variational Bayesian posterior inference often requires simplifying approximations such as mean-field parametrisation to ensure tractability. However, prior work has associated the variational mean-field approximation for Bayesian neural networks with underfitting in the case of small datasets or large model sizes. In this work, we show that invariances in the likelihood function of over-parametrised models contribute to this phenomenon because these invariances complicate the structure of the posterior by introducing discrete and/or continuous modes which cannot be well approximated by Gaussian mean-field distributions. In particular, we show that the mean-field approximation has an additional gap in the evidence lower bound compared to a purpose-built posterior that takes into account the known invariances. Importantly, this invariance gap is not constant; it vanishes as the approximation reverts to the prior. We proceed by first considering translation invariances in a linear model with a single data point in detail. We show that, while the true posterior can be constructed from a mean-field parametrisation, this is achieved only if the objective function takes into account the invariance gap. Then, we transfer our analysis of the linear model to neural networks. Our analysis provides a framework for future work to explore solutions to the invariance problem.

## 1 Introduction

Bayesian neural networks (BNNs) have several appealing advantages compared to deterministic neural networks (NN) such as improving generalization [38], capturing epistemic uncertainty [17], and providing a framework for continual learning methods [18, 23]. Unfortunately, reaping these theoretical benefits has so far been impeded [37]. In particular, several practical issues with variational Bayesian inference methods—which are the de-facto standard technique for scaling inference in BNNs to large datasets [15, 2]—have been identified to be partially responsible for a performance gap compared to deterministic NNs[16]. The most common variational approximation of the posterior is a product of independent Normal distributions, commonly referred to as the *mean-field* approximation. It has been observed that mean-field variational BNNs (VBNNs) suffer from severe underfitting for large models and small dataset sizes [13]. Recent work [5] showed that under certain assumptions such as an odd Lipschitz activation function and a finite dataset with bounded likelihood, the optimal mean-field approximation *collapses* to the prior as the NN width increases, *ignoring the data*.

---

[*]Correspondence to `kurler@amazon.com`.
[†]Work done while at AWS AI Labs.

36th Conference on Neural Information Processing Systems (NeurIPS 2022).

The goal of this work is to shed light on *why* the mean-field variational approximation collapses and to provide a new angle for future works to address this shortcoming. To this end, we study invariances in the likelihood function of *overparametrised models* and their detrimental effect on variational inference. For instance, it is well known that NNs exhibit several invariances *with respect to their parameters*, including node permutation invariance [19], sign-flip invariance (in the case of odd activation functions) [29, 3], scaling invariance (in the case of piece-wise linear activations) [26], and as we will show in Sec. 5, the parameter space of BNNs additionally has subspaces with *translation invariance*.

Invariance in the likelihood function does not necessarily pose a challenge if maximum likelihood point estimation through stochastic gradient descent is used, because convergence to any of the equivalent optima (resulting from the invariance) suffices for prediction. However, these invariances are detrimental to the variational Bayesian approach. As a first step to show this, we isolate the impact of the invariances in Sec. 3. To this end, we construct both a *mean-field* as well as an *invariance-abiding* approximation which explicitly models the invariances by integrating over all transformations that leave the likelihood invariant. Notably, both aforementioned posterior approximations are constructed from the same (mean-field) *likelihood* approximation. We then prove that, under the conditions outlined in Sec. 3, the mean-field approximation induces the same posterior predictive distribution as the invariance-abiding approximation. However, we also prove that the ELBO objective corresponding to these two approximations differs by the KL divergence between both posterior approximations; we refer to this difference as *invariance gap*. Importantly, the gap vanishes if both distributions are identical, which is the case if the mean-field approximation reverts to the prior.

We then demonstrate the detrimental effect of invariances in the likelihood function of an overparametrised Bayesian linear regression model in Sec. 4. This model is purposely selected as the canonical model exhibiting (only) *translation invariance*. We provide a detailed analysis of this model, including a tractable solution for the invariance-abiding approximation. It turns out that the optimal parameters w.r.t. the ELBO objective with the invariance-abiding distribution result in the true posterior. In contrast, posterior approximations with parameters that are optimal w.r.t. the mean-field ELBO revert to the prior as the number of dimensions increases.

Finally, we transfer our analysis of the linear model to VBNNs in Sec. 5, showing that subspaces in BNNs exist that exhibit translation invariance. Combined with a previous analysis of the node permutation invariance (cf. Kurle et al. [19] or App. E of the supplementary material), this provides the basis for future work to approximate the (generally intractable) invariance gap and thereby optimise for a tighter and favorable ELBO objective. We start in Sec. 2 by introducing the basic concepts of VBNNs and recent results on which we build our contribution.

## 2 Background

### 2.1 Variational Bayesian Neural Networks

**Neural network functional model.** Deep NNs are layered models that progressively transform their inputs in each layer. More formally, an $L$–layer NN computes algebraically

$$f(\mathbf{x}) = h_L\left(\mathbf{w}_{L,1}^{\mathsf{T}}\mathbf{z}_{L-1}\right), \quad z_{1,i} = h_1\left(\mathbf{w}_{1,i}^{\mathsf{T}}\mathbf{x}\right), \quad z_{l,i} = h_l\left(\mathbf{w}_{l,i}^{\mathsf{T}}\mathbf{z}_{l-1}\right),$$

where the $h_l : \mathbb{R} \to \mathbb{R}$ are monotonic transfer functions that introduce non-linearities. Such a network has $n_1 + n_2 + \cdots + n_L$ many nodes $z_{l,i}$, where $i$ indexes the layer-wise vectors $\mathbf{z}_l$. These nodes are each a weighted linear combination of either the input vector $\mathbf{x}$ (first hidden layer) or the value of the hidden units $\mathbf{z}_l$ (all other hidden layers and the output $f(\mathbf{x})$). We denote all learnable parameters of the model by the stacked weight vectors of each layer and node, $\mathbf{w} = [\mathbf{w}_{1,1}^{\mathsf{T}}, \ldots, \mathbf{w}_{L,n_L}^{\mathsf{T}}]^{\mathsf{T}}$.

**Variational Bayesian treatment.** If the weights and biases $\mathbf{w}$ are treated as random variables with a prior distribution $p(\mathbf{w})$, then the posterior $p(\mathbf{w} \,|\, \mathcal{D})$—induced by the dataset $\mathcal{D}$ through the likelihood function $\ell(\mathbf{w}; \mathcal{D}) := p(\mathcal{D} \,|\, \mathbf{w})$ that is defined via $f(\mathbf{x})$—is referred to as a Bayesian neural network (BNN). The variational Bayesian method approximates the posterior by a distribution $q_{\boldsymbol{\theta}}(\mathbf{w}) \approx p(\mathbf{w} \,|\, \mathcal{D})$ with variational parameters $\boldsymbol{\theta}$, casting inference as an optimization problem

$$q_{\boldsymbol{\theta}}^*(\mathbf{w}) = \underset{q_{\boldsymbol{\theta}} \in \mathcal{Q}}{\operatorname{argmin}} \ \mathrm{KL}\left[q_{\boldsymbol{\theta}} \,||\, p(\cdot \,|\, \mathcal{D})\right], \tag{1}$$

where $\text{KL}\left[q_{\boldsymbol{\theta}} \,||\, p(\cdot \,|\, \mathcal{D})\right] := \mathbb{E}_{\mathbf{w} \sim q_{\boldsymbol{\theta}}}\left[\ln q_{\boldsymbol{\theta}}(\mathbf{w}) - \ln p(\mathbf{w}|\mathcal{D})\right]$ and $\mathcal{Q}$ is a family of distributions over $\mathbf{w}$. The optimization of (1) is achieved by maximising a lower bound to the (log) model evidence $\ln p(\mathcal{D})$,

$$\mathcal{L}_{\text{ELBO}}\left(q_{\boldsymbol{\theta}}, \mathcal{D}\right) = \ln p(\mathcal{D}) - \text{KL}\left[q_{\boldsymbol{\theta}} \,||\, p(\cdot \,|\, \mathcal{D})\right] = \mathbb{E}_{\mathbf{w} \sim q_{\boldsymbol{\theta}}}\left[\ln \frac{\ell(\mathbf{w}; \mathcal{D})\, p(\mathbf{w})}{q_{\boldsymbol{\theta}}(\mathbf{w})}\right] \tag{2}$$
$$= \text{ELL}\left(q_{\boldsymbol{\theta}}, \,\mathcal{D}\right) - \text{KL}\left[q_{\boldsymbol{\theta}} \,||\, p\right],$$

where $\text{ELL}\left(q_{\boldsymbol{\theta}}, \,\mathcal{D}\right) := \mathbb{E}_{\mathbf{w} \sim q_{\boldsymbol{\theta}}}\left[\ln \ell(\mathbf{w}; \mathcal{D})\right]$ is the expected log-likelihood.

**Definition 1** (Mean-field variational BNN). A *mean-field* variational BNN (VBNN) is the minimizer of (1) where $\mathcal{Q}$ is the family of Gaussians with diagonal covariance matrix.

## 2.2 Data-related bound on the KL divergence

The term $\text{KL}\left[q_{\boldsymbol{\theta}} \,||\, p\right]$ in (2) admits a data-dependent upper bound that naturally occurs as a consequence of the finite information provided by a finite dataset in the presence of noise. This result has been shown in [5] and we recall the result for a Gaussian likelihood with homogeneous noise (for other likelihood functions, we refer to App. F in [5]).

Assume a VBNN with Gaussian observation noise defined through $y = f_{\mathbf{w}}(\mathbf{x}) + \epsilon$, with $\epsilon \sim \mathcal{N}(0, \sigma_y^2)$, an isotropic Gaussian prior $p(\mathbf{w}) = \mathcal{N}(\mathbf{w}; \mathbf{0}, \mathbf{I})$, and a mean-field variational approximation $q_{\boldsymbol{\theta}}(\mathbf{w}) = \mathcal{N}(\mathbf{w}; \mathbf{m}, \text{Diag}(\mathbf{v}))$. Define the NN output variance $\sigma_L^2\left(\mathbf{x}^{(n)}\right) := \mathbb{V}_{\mathbf{w} \sim p}\left[f(\mathbf{x}^{(n)}; \mathbf{w})\right]$ for some fixed input $\mathbf{x}^{(n)}$, where $L$ indicates the last layer. Then,

$$\text{KL}\left[q_{\boldsymbol{\theta}} \,||\, p\right] \le \text{ELL}\left(q^*, \mathcal{D}\right) - \text{ELL}\left(p, \mathcal{D}\right) = \sum_{n=1}^{N} \frac{\sigma_L^2\left(\mathbf{x}^{(n)}\right) + \left(y^{(n)}\right)^2}{2\sigma_y^2}, \tag{3}$$

where $q^*$ is a hypothetical approximation that predicts the data perfectly up to the noise variance $\sigma_y^2$ (see App. F for details). Prior work [5] has used the data-related bound to prove *that* the predictive distribution of mean-field BNNs converges to the prior predictive distribution if the network width is large and the activation function is odd, ultimately resulting in posterior collapse to the prior. In contrast, we address the question *why* the mean-field approximation cannot approximate the posterior by relating this data-related bound to invariances in neural network functions in the following section.

## 3 Invariance-abiding variational approximation

We develop a framework that enables modelling the invariances in the likelihood and understanding their impact on the ELBO objective. To achieve this, we approximate the likelihood by a Gaussian function with variational parameters and marginalise over all transformations to which the true likelihood is invariant. By taking the product with the prior, we construct an *invariance-abiding* posterior approximation $q_{\text{mix}}$ which can be related to a *mean-field* approximation $q_0$ that does not model invariances. We then describe conditions under which both approximations yield an identical posterior predictive, while the KL regularisation term of $q_0$ is lower bounded by the KL of $q_{\text{mix}}$.

**Variational likelihood and posterior approximations.** Non-identifiability implies that the posterior does not concentrate on a single set of parameters irrespective of the dataset size, because the same likelihood is assigned to different parameter values [12]. We model this invariance through transformations $t(\cdot, \mathbf{r})$ to which the likelihood $\ell(\mathbf{w}; \mathcal{D}) = p(\mathcal{D} \,|\, \mathbf{w})$ is invariant via variables $\mathbf{r}$:

$$\forall \mathbf{r} \sim p(\mathbf{r}) : \quad \ell(t(\mathbf{w}, \mathbf{r}); \mathcal{D})) = \ell(\mathbf{w}; \mathcal{D}). \tag{4}$$

From (4) it follows that the likelihood is also invariant w.r.t. the marginalisation

$$\ell\left(\mathbf{w}; \mathcal{D}\right) = \mathbb{E}_{\mathbf{r} \sim p}\left[\ell\left(t\left(\mathbf{w}, \mathbf{r}\right); \mathcal{D}\right)\right].$$

We assume that each of the equivalent parametrisations $\mathbf{w}' = t(\mathbf{w}, \mathbf{r})$ of the likelihood has the same probability a priori. For discontinuous transformations such as node permutations (see App. E), $p(\mathbf{r})$ is a uniform distribution over discrete variables indexing these transformations. For the continuous translation invariance (see Sec. 4), we model the uniform distribution as a Gaussian with infinite variance. It is conceivable that the likelihood can be constructed by marginalising over these transformations of a simpler function $\ell_0(\mathbf{w}; \mathcal{D})$ such that $\ell(\mathbf{w}; \mathcal{D}) = \mathbb{E}_{\mathbf{r} \sim p}\left[\ell_0(t(\mathbf{w}, \mathbf{r}); \mathcal{D})\right]$. For

instance, permutation invariance induces a factorial number of discontinuous modes that are each equivalent (cf. [19], App. E); and as we show in Sec. 4, the full covariance Gaussian likelihood and posterior of an over-parametrised Bayesian linear regression model can be constructed from a product of independent Gaussians. Curiously, it may be sufficient to approximate $\ell_0$ while taking into account the known invariances. To this end, we define a Gaussian *variational likelihood approximation* $g_0(\mathbf{w}; \boldsymbol{\theta}) \approx \ell_0(\mathbf{w}, \mathcal{D})$, with variational parameters $\boldsymbol{\theta}$. From this single mode approximation, we model an *invariance-abiding likelihood* approximation using the same parametrisation through

$$g_{\mathrm{mix}}(\mathbf{w}; \boldsymbol{\theta}) := \mathbb{E}_{\mathbf{r} \sim p} [g_0(t(\mathbf{w}, \mathbf{r}); \boldsymbol{\theta})] \approx \mathbb{E}_{\mathbf{r} \sim p} [\ell_0(t(\mathbf{w}, \mathbf{r}), \mathcal{D})] = \ell(\mathbf{w}; \mathcal{D}). \tag{5}$$

We consider mean-field Gaussians for the prior $p(\mathbf{w})$ and likelihood approximation $g_0(\mathbf{w}; \boldsymbol{\theta})$, where $\boldsymbol{\theta} = \{\mathbf{m}, \boldsymbol{\lambda}\}$ are means and variances of $g_0$. We then define a *mean-field posterior* as the product

$$q_0(\mathbf{w}; \boldsymbol{\theta}) := Z_0^{-1} \, p(\mathbf{w}) \cdot g_0(\mathbf{w}; \boldsymbol{\theta}), \quad Z_0 = \int p(\mathbf{w}) \cdot g_0(\mathbf{w}; \boldsymbol{\theta}) d\mathbf{w}. \tag{6a}$$

Similarly, we define an *invariance-abiding posterior* as the product of prior and invariant likelihood:

$$q_{\mathrm{mix}}(\mathbf{w}; \boldsymbol{\theta}) := Z_{\mathrm{mix}}^{-1} \, p(\mathbf{w}) \cdot g_{\mathrm{mix}}(\mathbf{w}; \boldsymbol{\theta}), \quad Z_{\mathrm{mix}} = \int p(\mathbf{w}) \cdot g_{\mathrm{mix}}(\mathbf{w}; \boldsymbol{\theta}) d\mathbf{w}, \tag{6b}$$

where $Z_0$ and $Z_{\mathrm{mix}}$ are normalisation constants. While $q_0(\mathbf{w}; \boldsymbol{\theta})$ is a mean-field approximation, the invariance-abiding approximation $q_{\mathrm{mix}}(\mathbf{w}; \boldsymbol{\theta})$ is a mixture with infinite continuous or finite discrete modes depending on the type of invariance. We will describe continuous translation invariance in Sec. 4; for the discrete node permutation invariance, see App. E of the supplementary material.

**Posterior predictive equivalence.** Next, we discuss conditions under which we can construct approximations $q_{\mathrm{mix}}(\mathbf{w}; \boldsymbol{\theta})$ and $q_0(\mathbf{w}; \boldsymbol{\theta})$ that yield the same predictive distribution. We first write the density $q_{\mathrm{mix}}$ as an expectation of product densities,

$$q_{\mathrm{mix}}(\mathbf{w}; \boldsymbol{\theta}) = Z_{\mathrm{mix}}^{-1} \, p(\mathbf{w}) \cdot \int p(\mathbf{r}) \, g_0(t(\mathbf{w}, \mathbf{r}); \boldsymbol{\theta}) d\mathbf{r} = \int p(\mathbf{r}) \frac{p(\mathbf{w}) \cdot g_0(t(\mathbf{w}, \mathbf{r}); \boldsymbol{\theta})}{Z_{\mathrm{mix}}} d\mathbf{r}.$$

Then, we assume that there exists a mapping $\mathbf{r}' = \varphi(\mathbf{r})$ such that

$$\forall \mathbf{r} \sim p(\mathbf{r}) : p(\mathbf{w}) \cdot g_0(t(\mathbf{w}, \mathbf{r}); \boldsymbol{\theta}) = p(t(\mathbf{w}, \varphi(\mathbf{r}))) \cdot g_0(t(\mathbf{w}, \varphi(\mathbf{r})); \boldsymbol{\theta}). \tag{7}$$

The condition in (7) allows us to exploit the invariance property of the likelihood (approximation) also for each product density $q_0(t(\mathbf{w}, \varphi(\mathbf{r})); \boldsymbol{\theta}) \propto p(t(\mathbf{w}, \varphi(\mathbf{r}))) \cdot g_0(t(\mathbf{w}, \varphi(\mathbf{r})); \boldsymbol{\theta})$, because then

$$q_{\mathrm{mix}}(\mathbf{w}; \boldsymbol{\theta}) = \int \frac{Z_0(\mathbf{r})}{Z_{\mathrm{mix}}} p(\mathbf{r}) \cdot q_0(t(\mathbf{w}, \varphi(\mathbf{r})); \boldsymbol{\theta}) d\mathbf{r},$$

where the normalisation constants are $Z_0(\mathbf{r}) = \int p(\mathbf{w}) \cdot g_0(t(\mathbf{w}, \varphi(\mathbf{r})); \boldsymbol{\theta}) d\mathbf{w}$, and $Z_{\mathrm{mix}}$ is defined in (6b). We show that the condition in (7) holds for translation and permutation invariance in App. C.

The second condition is that all invariance transformations $t(\cdot, \mathbf{r})$ must be volume-preserving, i.e.

$$\forall \mathbf{r} \sim p(\mathbf{r}) : \left| \det \frac{\partial t(\mathbf{w}, \mathbf{r})}{\partial \mathbf{w}} \right|^{-1} = 1. \tag{8}$$

Although (7) and (8) are fairly restricting, common invariances such as translation and permutation invariance fulfil these conditions for Gaussian priors and likelihood approximations (see App. C). With (7) and (8), we can show the posterior predictive equivalence (see Lemma 1 in App. C)

$$\mathbb{E}_{\mathbf{w} \sim q_{\mathrm{mix}}} [\ln p(\mathcal{D} \,|\, \mathbf{w})] = \mathbb{E}_{\mathbf{w} \sim q_0} [\ln p(\mathcal{D} \,|\, \mathbf{w})]. \tag{9}$$

**Invariance gap.** If the conditions in (7) and (8) are met, we also show (see Lemma 2 in App. C)

$$\mathcal{L}_{\mathrm{ELBO}} (q_0, \mathcal{D}) - \mathcal{L}_{\mathrm{ELBO}} (q_{\mathrm{mix}}, \mathcal{D}) = \mathrm{KL} [q_0 \,||\, p] - \mathrm{KL} [q_{\mathrm{mix}} \,||\, p] = \mathrm{KL} [q_0 \,||\, q_{\mathrm{mix}}]. \tag{10}$$

Interestingly, the gap between the two respective ELBO objectives is given exactly by the relative entropy between the mean-field and invariance-abiding approximation; we therefore refer to it as the *invariance gap*. We make the following observation about the detrimental effect of the invariances: when maximising the standard ELBO $\mathcal{L}_{\mathrm{ELBO}} (q_0, \mathcal{D})$ instead of the tighter objective $\mathcal{L}_{\mathrm{ELBO}} (q_{\mathrm{mix}}, \mathcal{D})$ w.r.t. the parameters of the variational *likelihood* approximation $g_0(\mathbf{w}; \boldsymbol{\theta})$ (used to construct both $q_0$ and $q_{\mathrm{mix}}$), we see that the former objective favors solutions where $q_0$ and $q_{\mathrm{mix}}$ coincide. This suboptimal solution is obtained if the mean-field posterior $q_0(\mathbf{w})$ and thus also the invariance-abiding posterior $q_{\mathrm{mix}}(\mathbf{w})$ revert to the prior. This is the case if $g_0(\mathbf{w})$ is uniform, because then $g_{\mathrm{mix}}(\mathbf{w})$ is uniform as well, and because both posteriors are constructed as the product of prior and likelihood approximation (cf. (6a)).

**Implication of data-related bound.** Since $\mathrm{KL}\left[q_0 \,\|\, p\right]$ is upper bounded by the best- and worst-case ELL as described in Sec. 2.2, the invariance gap is also bounded (using (10) and (3)):

$$\mathrm{KL}\left[q_0 \,\|\, q_{\mathrm{mix}}\right] \leq \mathrm{KL}\left[q_0 \,\|\, p\right] \leq \mathrm{ELL}\left(q^*, \mathcal{D}\right) - \mathrm{ELL}\left(p, \mathcal{D}\right). \tag{11}$$

Consequently, this bound puts a constraint on the achievable solutions to the maximisation of the ELBO objective w.r.t. the variational parameters of the mean-field approximation $q_0(\mathbf{w}; \boldsymbol{\theta})$. As discussed above, one specific parametrisation for which the invariance gap vanishes is the mean-field posterior collapse, $q_0(\mathbf{w}; \boldsymbol{\theta}) \approx p(\mathbf{w})$. However, to show that the invariance gap not only admits posterior collapse as a potential parametrisation but indeed incentivises it, we next tackle the question how the invariance gap behaves for non-collapsed approximations. One particularly relevant variational parametrisation is the optimal solution w.r.t. the ELBO objective with the invariance-abiding approximation. In order to tackle this question, we now consider a simple model for which the relevant distributions and the invariance gap can be computed exactly.

## 4 Translation invariance in linear models

We study an over-parametrised Bayesian linear regression model as the canonical model that exhibits *translation invariance*. The model serves as a useful tool to understand the detrimental effect of translation invariance since all interesting quantities can be computed analytically, incl. the mean-field and invariance-abiding distribution defined in (6) and the invariance gap from (10). We show how to lift results from this canonical model to the more general case of NNs in Sec. 5.

**Likelihood model.** Consider a linear model with $K$ latent variables $\mathbf{w} = [w_1, \ldots, w_K]^{\mathrm{T}}$ and assume that we have $N$ observations of variables $y$ given dependent inputs $\mathbf{x}$. To simplify the setting, we will assume that $\mathbf{x} = \mathbf{1}$ (see App. D for the more general case). We further assume that the observation depends only on the inner product with the inputs and additive Gaussian noise. That is, $y = \frac{1}{K}\mathbf{1}^{\mathrm{T}}\mathbf{w} + \epsilon, \quad \epsilon \sim \mathcal{N}\left(0, \sigma_y^2\right)$. Note that any change in $\mathbf{w}$ which leaves the sum of the elements unaffected does not change the likelihood. In general, we can model this translation invariance using a $K - 1$ dimensional vector $\boldsymbol{\Delta} \in \mathbb{R}^{K-1}$ and observing that

$$\mathbf{1}^{\mathrm{T}}\mathbf{w} = \mathbf{1}^{\mathrm{T}}\left(\mathbf{w} + \mathbf{B}\boldsymbol{\Delta}\right), \quad \mathbf{B} := \left[\begin{array}{c} \mathbf{I} \\ -\mathbf{1}^{\mathrm{T}} \end{array}\right], \tag{12}$$

since $\mathbf{1}^{\mathrm{T}}\mathbf{B}\boldsymbol{\Delta} = \mathbf{0}$. This over-parametrised model exhibits translation invariance $t(\mathbf{w}, \Delta) = \mathbf{w} - \mathbf{B}\Delta$.

**Prior.** We assume a Gaussian prior $p\left(\mathbf{w}\right) = \mathcal{N}\left(\mathbf{w}; \boldsymbol{\mu}, \boldsymbol{\Sigma}\right)$ with diagonal covariance $\boldsymbol{\Sigma} = \mathrm{Diag}\left(\boldsymbol{\sigma}^2\right)$, where we consider $\boldsymbol{\sigma}^2 = K \cdot \sigma_0^2 \cdot \mathbf{1}$ proportional to the number of dimensions $K$, such that the predictive variance $\mathbb{V}\left[\frac{1}{K}\sum_k w_k + \epsilon\right]$ is constant w.r.t. $K$. Another way to view this is to model a prior over parameters that induces the same prior over functions for different $K$.

**Posterior.** The posterior of this Gaussian linear model can be computed as (see App. D.3)

$$p(\mathbf{w} \,|\, \mathbf{y}) = \mathcal{N}\left(\mathbf{w}; \mathbf{m}_p^*, \mathbf{V}_p^*\right), \quad \mathbf{V}_p^* = \left(\frac{N}{K^2\sigma_y^2}\mathbf{1}\mathbf{1}^{\mathrm{T}} + \boldsymbol{\Sigma}^{-1}\right)^{-1}, \quad \mathbf{m}_p^* = \mathbf{V}_p^*\frac{\sum_i y_i}{K\sigma_y^2}\mathbf{1}. \tag{13}$$

As can be seen, the resulting posterior has full covariance with a diagonal plus rank-1 matrix. We will now show that we can construct this posterior from the prior and a mean-field Gaussian approximation of the likelihood by marginalising over all translations $\Delta \sim p(\Delta)$ to which the likelihood is invariant.

### 4.1 Mean-field parametrisation of the likelihood function

Next, we construct the mean-field and invariance-abiding posterior approximations defined in (6) from the prior defined above and the mean-field likelihood approximation with locations $\mathbf{m}$ and variances $\boldsymbol{\lambda}$. We model that the likelihood is translation invariant by computing the marginal from (5) and considering $p(\Delta) = \mathcal{N}\left(\boldsymbol{\Delta}; \mathbf{0}, \beta^2\mathbf{I}\right)$ with $\beta \to \infty$ as the uniform distribution over all translations:

$$g_{\mathrm{mix}}\left(\mathbf{w}; \boldsymbol{\theta}\right) = \int g_0\left(t(\mathbf{w}, \Delta); \boldsymbol{\theta}\right) \cdot p\left(\boldsymbol{\Delta}\right) d\boldsymbol{\Delta}$$

$$= \lim_{\beta \to \infty} \int \mathcal{N}\left(\mathbf{w}; \mathbf{m} + \mathbf{B}\boldsymbol{\Delta}, \mathrm{Diag}\left(\boldsymbol{\lambda}\right)\right) \cdot \mathcal{N}\left(\boldsymbol{\Delta}; \mathbf{0}, \beta^2\mathbf{I}\right) d\boldsymbol{\Delta} =: \mathcal{N}\left(\mathbf{w}; \mathbf{m}_{\mathrm{mix}}, \mathbf{V}_{\mathrm{mix}}\right).$$

Writing the uniform distribution as the limiting case of a Gaussian with infinite variance allows us to compute the integral analytically. As shown in App. D, the resulting function is a multivariate Gaussian with a degenerate rank-1 covariance matrix and the same location parameter as $g_0(\mathbf{w})$:

$$\mathbf{m}_{\mathrm{mix}} = \mathbf{m}, \quad \mathbf{V}_{\mathrm{mix}}^{-1} = \frac{1}{\mathbf{1}^{\mathrm{T}}\boldsymbol{\lambda}}\mathbf{1}\mathbf{1}^{\mathrm{T}}. \tag{14}$$

However, the invariance-abiding posterior is non-degenerate, and it can be computed efficiently as

$$q_{\mathrm{mix}}\left(\mathbf{w}; \boldsymbol{\theta}\right) = \mathcal{N}\left(\mathbf{w}; \boldsymbol{\mu} + \frac{\mathbf{1}^{\mathrm{T}}\left(\mathbf{m} - \boldsymbol{\mu}\right)}{\mathbf{1}^{\mathrm{T}}\left(\boldsymbol{\lambda} + \boldsymbol{\sigma}^2\right)}\boldsymbol{\sigma}^2, \mathrm{Diag}\left(\boldsymbol{\sigma}^2\right) - \frac{1}{\mathbf{1}^{\mathrm{T}}\left(\boldsymbol{\lambda} + \boldsymbol{\sigma}^2\right)} \cdot \boldsymbol{\sigma}^2\left(\boldsymbol{\sigma}^2\right)^{\mathrm{T}}\right). \tag{15}$$

As can be seen, this covariance matrix is not diagonal; it has an additive rank-1 term that vanishes as $\lambda \to \infty$. Interestingly, the location parameter of $q_{\mathrm{mix}}$ is translated from the *prior* location $\boldsymbol{\mu}$ along the direction of the *prior* variance vector $\boldsymbol{\sigma}^2$. This is because $g_{\mathrm{mix}}$ is uniform along the $K-1$ dimensions hyper-plane determined by its normal vector $\mathbf{1}$. Taking the Gaussian product then translates the location in the direction $\mathrm{Diag}(\boldsymbol{\sigma}^2)\mathbf{1} = \boldsymbol{\sigma}^2$. Note also that the invariance-abiding posterior can be written in the same form as the true posterior in (13) (see App. D.3). The optimal invariance-abiding posterior is thus the true posterior. We visualise the two respective posterior approximations and the corresponding likelihood in Fig. 3 of App. D with two different parametrisations (cf. Sec. 4.3).

## 4.2   Invariance gap

To quantify the detrimental effect of the translation invariance in the considered linear model we now compute the invariance gap in (11). Note again that we assume a scaled standard normal prior with variance $\boldsymbol{\sigma}^2 = K\sigma_0^2 \cdot \mathbf{1}$. We simplify the form of the KL divergence by assuming that all variances and means take the same value in the likelihood (and thus also in the posterior) approximation, i.e.

$$\forall k : \lambda_k = \hat{\lambda}, \, m_k = \hat{m}, \, \sigma_k = \hat{\sigma}, \, \mu_k = \hat{\mu}.$$

This choice is motivated by the fact that the posterior approximation is a function of the sum $\mathbf{1}^{\mathrm{T}}\boldsymbol{\lambda}$ only (cf. (15)), and, hence, it makes no difference for $q_{\mathrm{mix}}$. However, since the prior variance is proportional to $\mathbf{1}$, the Gaussian likelihood resulting in the highest ELBO for the approximation $q_0$ also has a variance vector proportional to $\mathbf{1}$. Using this assumption, the invariance gap is

$$\mathrm{KL}\left[q_0 \,||\, q_{\mathrm{mix}}\right] = \frac{K-1}{2}\left[\ln\left(\frac{\hat{\sigma}^2 + \hat{\lambda}}{\hat{\lambda}}\right) + \frac{\hat{\lambda}}{\hat{\sigma}^2 + \hat{\lambda}} - 1\right]. \tag{16}$$

This is a convex function in $\phi = \frac{\hat{\sigma}^2 + \hat{\lambda}}{\hat{\lambda}}$ with a minimum at $\phi = 1$; it is minimised as $\hat{\lambda} \to \infty$. It is however not evident whether the gap has a large magnitude compared to the rest of the regularisation term and over-regularises in practice. In the next section, we therefore analyse this term at the optima for the mean-field and the invariance-abiding approximations defined in (17). The corresponding invariance gaps are visualised in Fig. 1 where it can be seen that the invariance gap grows linearly when using the optimal parameters of the invariance-abiding distribution and the unattainable data-related bound is reached quickly, thus preventing this optimum if (17b) is optimised instead of (17a).

## 4.3   Optimal mean-field and invariance-abiding parametrisations

To better understand the detrimental effect of the invariance gap, we now analyse the ELL and KL terms of the ELBO objective for the mean-field and invariance-abiding variational approximations, respectively. We compare these terms for both distributions with the parameters optimized for both posterior distributions, giving $2 \times 2$ combinations. We denote the optimal parameters w.r.t. the invariance-abiding approximation and the mean-field approximation, respectively, as

$$\boldsymbol{\theta}_{\mathrm{mix}}^* = \underset{\boldsymbol{\theta}}{\mathrm{argmax}}\ \mathcal{L}_{\mathrm{ELBO}}\left(q_{\mathrm{mix}}\left(\cdot\,;\boldsymbol{\theta}\right), \mathcal{D}\right), \tag{17a}$$

$$\boldsymbol{\theta}_0^* = \underset{\boldsymbol{\theta}}{\mathrm{argmax}}\ \mathcal{L}_{\mathrm{ELBO}}\left(q_0\left(\cdot\,;\boldsymbol{\theta}\right), \mathcal{D}\right). \tag{17b}$$

Perhaps the simplest way compute these optimal parameters is to notice that $q_{\mathrm{mix}}$ in (15) can be written in the same form as the true posterior and then simply read out the optimal parameters. This is shown in App. D.3; the resulting optimal variational parameters are

$$g_0(\mathbf{w}\,;\boldsymbol{\theta}_{\mathrm{mix}}^*) = \mathcal{N}\left(\mathbf{w}; \mathbf{m}_{\mathrm{mix}}^*, \mathrm{Diag}\left(\boldsymbol{\lambda}_{\mathrm{mix}}^*\right)\right), \quad \mathbf{m}_{\mathrm{mix}}^* = \frac{1}{N}\sum_{n=1}^{N}y^{(n)}\cdot\mathbf{1}, \quad \boldsymbol{\lambda}_{\mathrm{mix}}^* = \frac{K\sigma_y^2}{N}\cdot\mathbf{1}. \tag{18}$$

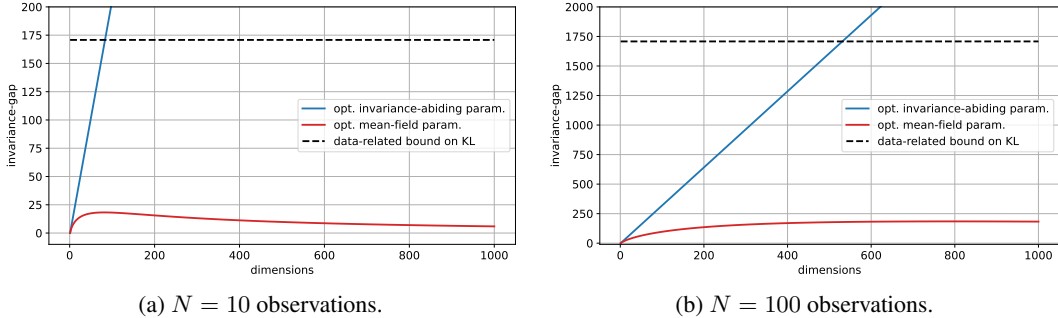

(a) $N = 10$ observations.          (b) $N = 100$ observations.

Figure 1: Invariance gap evaluated at different optima (cf. (17)), i.e. $\mathrm{KL}\left[q_0(\cdot;\boldsymbol{\theta}^*_{\mathrm{mix}})\,\|\,q_{\mathrm{mix}}(\cdot;\boldsymbol{\theta}^*_{\mathrm{mix}})\right]$ and $\mathrm{KL}\left[q_0(\cdot;\boldsymbol{\theta}^*_0)\,\|\,q_{\mathrm{mix}}(\cdot;\boldsymbol{\theta}^*_0)\right]$. Prior variances are $\sigma^2 = K \cdot \mathbf{1}$, where $K$ are the dimensions; the noise variance is $\sigma^2_y = (2\pi\mathrm{e})^{-1}$; all observations $y = 1$ and $\mathbf{x} = \mathbf{1}$ are identical. As $K$ increases, the invariance gap vanishes in case of the optimal parameters $\boldsymbol{\theta}^*_0$. In contrast, the gap for $\boldsymbol{\theta}^*_{\mathrm{mix}}$, which induces the true posterior predictive, grows linearly. As the data-related bound (cf. Sec. 2.2) can not be exceeded by the optimal parameters, $\boldsymbol{\theta}^*_0$ can not coincide with the optimal $\boldsymbol{\theta}^*_{\mathrm{mix}}$.

The optimal parameters for the mean-field *posterior* can also be computed analytically, since the Gaussian mean-field distribution that minimises the KL divergence to the multivariate Gaussian true posterior is known [34]. The resulting optimal parameters of the mean-field *likelihood* are

$$g_0(\mathbf{w}\,;\,\boldsymbol{\theta}^*_0) = \mathcal{N}\left(\mathbf{w}; \mathbf{m}^*_0, \mathrm{Diag}\left(\boldsymbol{\lambda}^*_0\right)\right), \quad \mathbf{m}^*_0 = \mathbf{m}^*_{\mathrm{mix}}, \quad \boldsymbol{\lambda}^*_0 = \frac{K^2\sigma^2_y}{N}\,\mathbf{1}. \tag{19}$$

As can be seen, the two respective optimal parameters differ by the factor $K$ in the likelihood variance. We then construct the two respective posterior approximations $q_0$ and $q_{\mathrm{mix}}$ with these two optimal parameters and compare the $2 \times 2$ combinations in Fig. 2, where we visualise the ELBO terms.

Note again that we model prior variances $\boldsymbol{\sigma}^2 = K\sigma^2_0 \cdot \mathbf{1}$ such that prior and posterior over functions are identical for any $K$. Due to this choice and since $q_{\mathrm{mix}}(\mathbf{w};\boldsymbol{\theta}^*_{\mathrm{mix}})$ approximates the true posterior exactly, it does not suffer from over-parametrisation. This can be seen by the loss terms in Fig. 2 being constant in $K$. In contrast, since $\boldsymbol{\lambda}^*_0$ depends quadratically on $K$ and $\boldsymbol{\sigma}^2$ only linearly, $q_{\mathrm{mix}}(\mathbf{w};\boldsymbol{\theta}^*_0)$ *collapses* to the prior as $K \to \infty$. This can be seen e.g. by the shrinking KL regularisation (Fig. 2a) and ELL (in 2b) term. As a consequence, and in line with Coker et al. [5], the posterior predictive variance of the optimal mean-field approximation reverts to the prior predictive variance (Fig. 2d).

We have shown that—in the simplified setting of a linear model with dependent observations—the consequence of not handling translation invariance is indeed posterior collapse as $K \to \infty$, while modelling the invariance coincides with the true posterior. An important observation and key takeaway is that, while we need to optimise for (17a), the mean-field posterior approximation $q_0(\mathbf{w};\boldsymbol{\theta}^*_{\mathrm{mix}})$ is sufficient for prediction. Indeed, for the linear model, we could compute the invariance gap exactly and thereby correct the ELBO objective for the invariances of this model. For more complex non-linear models such as BNNs, this section may serve as a direction for approximating the invariance gap. To this end, the next section describes the layer-wise translation invariance exhibited by BNNs.

## 5 Translation invariance in Bayesian neural networks

Let us now go back to the general NN model in Sec. 2.1. In order to describe the set of all invariances, note that for each node $z_{l,j}$ in layer $l$ (or the output node $y$), there is a subspace that keeps the value $z_{l,j}$ (or $y$) invariant when using the translation-invariance model in (12), because the value itself only depends on the *actual* activation $\mathbf{z}_{l-1}$ (or the input $\mathbf{x}$).

More formally, the following equations model all translation invariances of a NN at a data point $\mathbf{x}$:

$$\forall \boldsymbol{\Delta}_{L,1}: \quad f(\mathbf{x}) = h_L\left(\mathbf{w}^\mathrm{T}_{L,1}\mathbf{z}_{L-1}\right) = h_L\left(\left(\mathbf{w}_{L,1} + \mathbf{B}_{\mathbf{z}_{L-1}}\boldsymbol{\Delta}_{L,1}\right)^\mathrm{T}\mathbf{z}_{L-1}\right), \tag{20}$$

$$\forall j \in \{1,\dots,n_l\}, \boldsymbol{\Delta}_{l,j}: \quad z_{l,j} = h_l\left(\mathbf{w}^\mathrm{T}_{l,j}\mathbf{z}_{l-1}\right) h_l\left(\left(\mathbf{w}_{l,j} + \mathbf{B}_{\mathbf{z}_{l-1}}\boldsymbol{\Delta}_{l,j}\right)^\mathrm{T}\mathbf{z}_{l-1}\right), \tag{21}$$

$$\forall j \in \{1,\dots,n_1\}, \boldsymbol{\Delta}_{1,j}: \quad z_{1,j} = h_1\left(\mathbf{w}^\mathrm{T}_{1,j}\mathbf{x}\right) = h_1\left(\left(\mathbf{w}_{1,j} + \mathbf{B}_{\mathbf{x}}\boldsymbol{\Delta}_{1,j}\right)^\mathrm{T}\mathbf{x}\right), \tag{22}$$

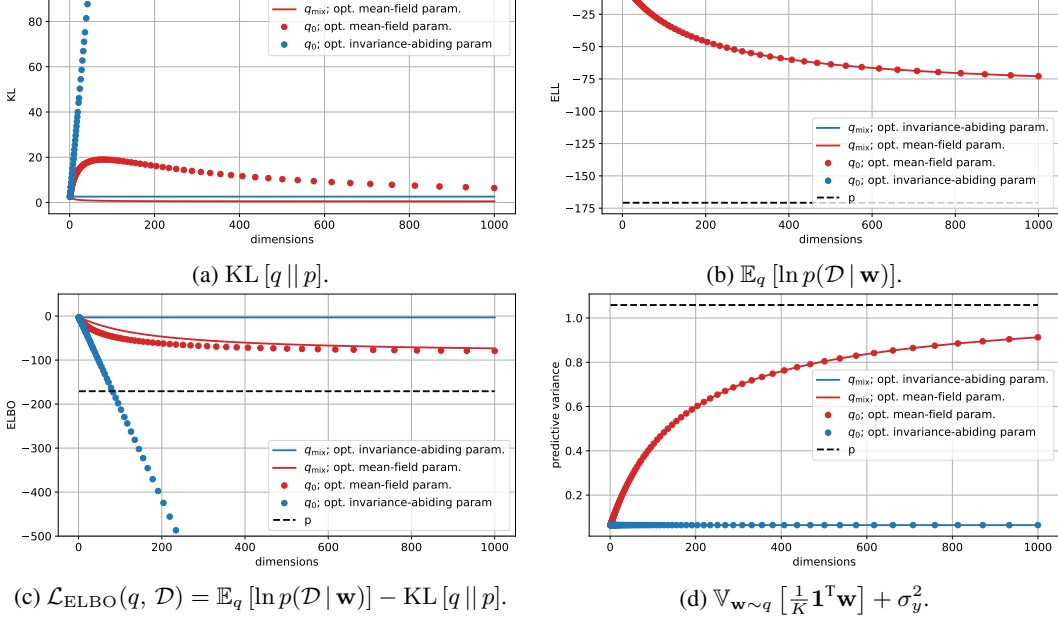

(a) KL $[q \,||\, p]$.

(b) $\mathbb{E}_q\left[\ln p(\mathcal{D} \,|\, \mathbf{w})\right]$.

(c) $\mathcal{L}_{\mathrm{ELBO}}(q, \mathcal{D}) = \mathbb{E}_q\left[\ln p(\mathcal{D} \,|\, \mathbf{w})\right] - \mathrm{KL}\left[q \,||\, p\right]$.

(d) $\mathbb{V}_{\mathbf{w} \sim q}\left[\frac{1}{K}\mathbf{1}^{\mathrm{T}}\mathbf{w}\right] + \sigma_y^2$.

Figure 2: ELBO loss terms and predictive variance for different posterior approximations. Lines/dots indicate the invariance-abiding/mean-field posterior, respectively. Red/blue indicates the optimal invariance-abiding/mean-field parameters (cf. (17)). Prior variance is $\boldsymbol{\sigma}^2 = K \cdot \mathbf{1}$, and $\sigma_y^2 = 1/\left(2\pi \mathrm{e}\right)$; we used $N = 10$ identical observations with value $y = 1$ and inputs $\mathbf{x} = \mathbf{1}$. Notably, both posterior approximations yield the same predictive distribution as can be seen by the ELL (b) and predictive variance (d). The variance of the mean-field approximation reverts to the prior variance as $K \to \infty$.

where the layer index $l$ ranges from $2, \ldots, L - 1$ and $\mathbf{B_z}$ is given by

$$\mathbf{B_z} := \left[\begin{array}{ccc} & \mathbf{I} & \\ -\frac{z_1}{z_k} & \cdots & -\frac{z_{k-1}}{z_k} \end{array}\right].$$

In order to efficiently compute the invariance-abiding likelihood $g_{\mathrm{mix}}$, note that it also decomposes over the NN layers and the associated parameters as follows: For the $n_1$ nodes in the first layer we have

$$q_{\mathrm{mix}}(\mathbf{w}_{1,j}) = \mathbb{E}_{\mathbf{x}}\left[\mathcal{N}\left(\mathbf{w}_{1,j}; \boldsymbol{\mu} + \frac{\mathbf{x}^{\mathrm{T}}\left(\mathbf{m} - \boldsymbol{\mu}\right)}{\mathbf{x}^{\mathrm{T}}\left(\mathbf{V} + \boldsymbol{\Sigma}\right)\mathbf{x}}\left(\boldsymbol{\Sigma}\mathbf{x}\right), \boldsymbol{\Sigma} - \frac{1}{\mathbf{x}^{\mathrm{T}}\left(\mathbf{V} + \boldsymbol{\Sigma}\right)\mathbf{x}}\left(\boldsymbol{\Sigma}\mathbf{x}\right)\left(\boldsymbol{\Sigma}\mathbf{x}\right)^{\mathrm{T}}\right)\right], \quad (23)$$

$$P(z_{1,j}) = \mathbb{E}_{\mathbf{x}}\left[\mathbb{E}_{\mathbf{w} \sim q_{\mathrm{mix}}(\mathbf{w}_{1,j})}\left[h_1\left(\mathbf{w}^{\mathrm{T}}\mathbf{x}\right)\right]\right], \quad (24)$$

where the outer expectation is taken over $\mathbf{x} \sim p(\mathcal{D})$ and $p(\mathcal{D})$ is the empirical distribution over the training set $\mathcal{D}$, and where $\mathbf{m}, \mathbf{V}, \boldsymbol{\mu}$ and $\boldsymbol{\Sigma}$ are constrained to the parts of the overall weight vector that correspond to $\mathbf{w}_{1,j}$. Now, since the invariance-abiding mechanism for nodes in layer $l$ only depends on the *value* $\mathbf{z}_{l-1}$ of the hidden units in layer $n - 1$ (see (20) and (21)), we have

$$q_{\mathrm{mix}}(\mathbf{w}_{l,j}) = \mathbb{E}_{\mathbf{z}}\left[\mathcal{N}\left(\mathbf{w}_{l,j}; \boldsymbol{\mu} + \frac{\mathbf{z}^{\mathrm{T}}\left(\mathbf{m} - \boldsymbol{\mu}\right)}{\mathbf{z}^{\mathrm{T}}\left(\mathbf{V} + \boldsymbol{\Sigma}\right)\mathbf{z}}\left(\boldsymbol{\Sigma}\mathbf{z}\right), \boldsymbol{\Sigma} - \frac{1}{\mathbf{z}^{\mathrm{T}}\left(\mathbf{V} + \boldsymbol{\Sigma}\right)\mathbf{z}}\left(\boldsymbol{\Sigma}\mathbf{z}\right)\left(\boldsymbol{\Sigma}\mathbf{z}\right)^{\mathrm{T}}\right)\right], \quad (25)$$

$$P(z_{l,j}) = \mathbb{E}_{\mathbf{z}}\left[\mathbb{E}_{\mathbf{w} \sim q_{\mathrm{mix}}(\mathbf{w}_{l,j})}\left[h_l\left(\mathbf{w}^{\mathrm{T}}\mathbf{z}_{l-1}\right)\right]\right], \quad (26)$$

where again $\mathbf{m}, \mathbf{V}, \boldsymbol{\mu}$ and $\boldsymbol{\Sigma}$ are constrained to the parts of the overall weight vector that correspond to $\mathbf{w}_{l,j}$ and the expectation over $\mathbf{z}$ is taken with respect to $\mathbf{z} \sim P(\mathbf{z}_{l-1})$.

Thus, the invariance-abiding approximation could e.g. be computed with a layer-by-layer iterative optimisation: First, given data $\mathbf{x} \sim p(\mathcal{D})$ and prior $p(\mathbf{w})$, use (23) and point estimates for all weight vectors in later layers to optimise the mean $\mathbf{m}$ and diagonal covariance $\mathbf{V}$ of the weights in the first layer, i.e., $\{\mathbf{w}_{1,j} | j = 1, \ldots, n_1\}$. Then, we can use (24) to generate $P$ samples $\mathbf{z}_{1,n}$ of the activations of the first layer under the (now fitted) optimal $q_{\mathrm{mix}}(\{\mathbf{w}_{1,j}\})$. Next, for each of these samples $\mathbf{z}_{1,n}$, we can use (25) and point estimates for all weight vectors in later layers to optimise the mean $\mathbf{m}$ and diagonal covariance $\mathbf{V}$ of the weights in the second layer and average them for the optimal $q_{\mathrm{mix}}$ of the weights in the second layer. Finally, we can use (26) to generate samples $\mathbf{z}_{2,i}$ for the second layer.

This process is repeated until the last layer, and iterated until $q_{\mathrm{mix}}(\mathbf{w})$ converges. The complexity of this procedure is no larger than optimising the weight matrices of a single layer, because all previous layers are fully characterised by the distribution of the latent activations $\mathbf{z}_l$ of the hidden layers.

## 6 Related work

Poor empirical performance of VBNNs has been reported in several previous works, e.g., [37, 16, 33]. For instance, it has been shown that single-layer mean-field VBNNs with ReLU activations can not have large predictive uncertainty between regions of low uncertainty [10]. In contrast, Farquhar et al. [8] argue that mean-field approximations are expressive enough for deep BNNs; they prove a universality result that the predictive distribution of mean-field approximations of BNNs with at least 2 layers of hidden units *can* approximate any true posterior distribution over function values arbitrarily closely. This result seems in conflict with our work and [5], who show that the predictive distribution of mean-field VBNNs reverts to the prior predictive distribution as the network width increases. The discrepancy between these two results is resolved by noting that Farquhar et al. [8] only shows that the expressive power of the model is large enough but not that the approximate inference algorithms will converge to this solution. Our work sheds light on why the mean-field approximation fails to approximate expressive posteriors via the invariance gap in the ELBO. Our framework to model the invariances is similar to and extends previous preliminary works [19, 22].

Surprisingly, restricting the parametrisation of the variational posterior results in competitive or even better performance [30, 21, 31, 7]. For example, Dusenberry et al. [7] proposes a variational approximation only for rank-1 factors, resulting in inference of a lower-dimensional subspace. The outer product of these low-rank factors perturb a weight matrix that is treated deterministically through maximum a posteriori (MAP) estimation. This approach avoids the invariance problem associated with variational Bayesian inference since the MAP estimate is not impeded by this problem and the subspace of the variational weights do not possess the same invariances.

Other attempts have proposed to approximate the posterior predictive directly, thereby circumventing the non-identifiability issue [28, 20, 36]. While these approaches have shown promising empirical performance, estimating the KL regularisation term in function space is more complicated and can even be ill-defined due to regions of zero prior probability mass [4]. In a similar vein, previous work also proposed to map the BNN prior to a Gaussian process (functional) prior [9, 32]. Another promising direction to circumvent invariance in layered models are deep kernel processes, in which Gram matrices are progressively transformed by nonlinear kernel functions [1]. The Gram matrices, which are treated as the random variables, are invariant to permutations/rotations of the weights/features.

More broadly, (non-)identifiability of parameters and latent variables in probabilistic models is a widely-studied topic both from frequentist [27] and Bayesian perspectives [6, 25, 11, 24]. In a recent example, Wang et al. [35] attributed posterior collapse in the context of variational auto-encoders to non-identifiability of latent variables.

## 7 Conclusion

We have associated the posterior collapse phenomenon of mean-field VBNNs with invariances in the likelihood function, in particular translation invariance. While the invariance does not affect the predictive distribution, the approximations of the posterior—which abide and do not abide the invariance—differ in the KL regularisation term and consequently in the tightness of the ELBO objective. We proved that the objectives of the two approximations differ by the relative entropy (KL) between the standard mean-field approximation and the invariance-abiding distribution. We related this to a data-related bound on the KL regularisation, which prevents fits for which the gap is large.

We studied over-parametrised Bayesian linear regression as the canonical model that exhibits *translation invariance* and for which the relevant terms can be computed exactly. A detailed analysis of this model confirms our hypothesis that the invariance leads to a significant additional gap in the ELBO objective compared to approximations that model the invariance. It is this very gap which prevents mean-field approximation to achieve the same fit as an invariance-abiding approximation and instead leads to a collapse of the posterior variances to the prior variance.

While we can compute the invariance gap for the over-parametrised linear model, we have not yet identified a computationally efficient procedure for layered models or for other invariances. However, our work provides the mathematical tools to address over-regularisation due to invariances. Future work will focus on approximations of the invariance gap in order to correct the ELBO objective for translation and permutation invariances in general neural network functions.

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
