}} (\mathbf{m} - \boldsymbol{\mu})}{\mathbf{x}^{\mathrm{T}} (\mathbf{V} + \boldsymbol{\Sigma}) \mathbf{x}} (\boldsymbol{\Sigma} \mathbf{x}), \boldsymbol{\Sigma} - \frac{1}{\mathbf{x}^{\mathrm{T}} (\mathbf{V} + \boldsymbol{\Sigma}) \mathbf{x}} (\boldsymbol{\Sigma} \mathbf{x}) (\boldsymbol{\Sigma} \mathbf{x})^{\mathrm{T}} \right) \right], \quad (23)$$

$$P(z_{1,j}) = \mathbb{E}_{\mathbf{x}} \left[ \mathbb{E}_{\mathbf{w} \sim q_{\mathrm{mix}}(\mathbf{w}_{1,j})} \left[ h_1 \left( \mathbf{w}^{\mathrm{T}} \mathbf{x} \right) \right] \right], \quad (24)$$

where the outer expectation is taken over $\mathbf{x} \sim p(\mathcal{D})$ and $p(\mathcal{D})$ is the empirical distribution over the training set $\mathcal{D}$, and where $\mathbf{m}$, $\mathbf{V}$, $\boldsymbol{\mu}$ and $\boldsymbol{\Sigma}$ are constrained to the parts of the overall weight vector that correspond to $\mathbf{w}_{1,j}$. Now, since the invariance-abiding mechanism for nodes in layer $l$ only depends on the *value* $\mathbf{z}_{l-1}$ of the hidden units in layer $n - 1$ (see (20) and (21)), we have

$$q_{\mathrm{mix}}(\mathbf{w}_{l,j}) = \mathbb{E}_{\mathbf{z}} \left[ \mathcal{N} \left( \mathbf{w}_{l,j}; \boldsymbol{\mu} + \frac{\mathbf{z}^{\mathrm{T}} (\mathbf{m} - \boldsymbol{\mu})}{\mathbf{z}^{\mathrm{T}} (\mathbf{V} + \boldsymbol{\Sigma}) \mathbf{z}} (\boldsymbol{\Sigma} \mathbf{z}), \boldsymbol{\Sigma} - \frac{1}{\mathbf{z}^{\mathrm{T}} (\mathbf{V} + \boldsymbol{\Sigma}) \mathbf{z}} (\boldsymbol{\Sigma} \mathbf{z}) (\boldsymbol{\Sigma} \mathbf{z})^{\

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

## A  Notation

We denote matrices and vectors with bold upper- and lower-case letters, e.g. $\mathbf{a}$ and $\mathbf{A}$. In order to address an element of a vector or matrix, we will use non-bold letters, e.g. $\mathbf{x} = [x_1, x_2, \ldots, x_N]^{\mathrm{T}}$. We will use the superscript $^{\mathrm{T}}$ to denote a transpose of a vector and $|\mathbf{A}|$ to denote the determinant of the matrix $\mathbf{A}$. We use $\mathrm{Diag}(\mathbf{a})$ to denote the diagonal matrix constructed from a vector, and $\mathrm{diag}(\mathbf{A})$ to denote the diagonal vector of a matrix. We use the notation $\mathcal{N}(\mathbf{x}; \mathbf{m}, \mathbf{V})$ to denote the density of the Gaussian distribution given by

$$\mathcal{N}(\mathbf{x}; \boldsymbol{\mu}, \boldsymbol{\Sigma}) := |2\pi\boldsymbol{\Sigma}|^{-\frac{1}{2}} \cdot \exp\left(-\frac{1}{2}(\mathbf{x} - \boldsymbol{\mu})^{\mathrm{T}} \boldsymbol{\Sigma}^{-1} (\mathbf{x} - \boldsymbol{\mu})\right).$$

An alternative representation of the Gaussian distribution is in terms of its canonical parameters $\boldsymbol{\eta} = \boldsymbol{\Sigma}^{-1}\boldsymbol{\mu}$ and $\boldsymbol{\Lambda} = \boldsymbol{\Sigma}^{-1}$:

$$\mathcal{G}(\mathbf{x}; \boldsymbol{\eta}, \boldsymbol{\Lambda}) := \exp\left(\mathbf{x}^{\mathrm{T}}\boldsymbol{\eta} - \frac{1}{2}\mathbf{x}^{\mathrm{T}}\boldsymbol{\Lambda}\mathbf{x} - \frac{1}{2}\boldsymbol{\eta}^{\mathrm{T}}\boldsymbol{\Lambda}^{-1}\boldsymbol{\eta} - \frac{1}{2}\ln\left(|2\pi\boldsymbol{\Lambda}^{-1}|\right)\right).$$

Note that $\mathcal{G}(\mathbf{x}; \boldsymbol{\eta}_1, \boldsymbol{\Lambda}_1) \cdot \mathcal{G}(\mathbf{x}; \boldsymbol{\eta}_2, \boldsymbol{\Lambda}_2) \propto \mathcal{G}(\mathbf{x}; \boldsymbol{\eta}_1 + \boldsymbol{\eta}_2, \boldsymbol{\Lambda}_1 + \boldsymbol{\Lambda}_2)$ which renders this canonical parameterisation very useful when considering the multiplication of Gaussian densities. Furthermore, we use $\mathbb{E}_{\mathbf{w} \sim p}[f(\mathbf{w})]$ to denote the expectation of the function $f(\cdot)$ when $\mathbf{w}$ is drawn from the

distribution $p$. Also, we use $\mathbb{V}_{\mathbf{w}\sim p}\left[f(\mathbf{w})\right]$ to denote the variance of the function $f(\cdot)$ when $\mathbf{w}$ is drawn from the distribution $p$ defined by

$$\mathbb{V}_{\mathbf{w}\sim p}\left[f(\mathbf{w})\right] := \mathbb{E}_{\mathbf{w}\sim p}\left[\left(f(\mathbf{w}) - \mathbb{E}_{\mathbf{w}'\sim p}\left[f(\mathbf{w}')\right]\right)^2\right] .$$

## B   Convolution of Gaussian Measures

**Theorem 1** (Convolution of Gaussian Measures). *For any $\mathbf{A} \in \mathbb{R}^{n\times k}$ and $\mathbf{b} \in \mathbb{R}^n$ it holds that*

$$p\left(\mathbf{x}|\boldsymbol{\theta}\right) = \mathcal{N}\left(\mathbf{x}; \mathbf{A}\boldsymbol{\theta} + \mathbf{b}, \mathbf{V}\right) , \; p\left(\boldsymbol{\theta}\right) = \mathcal{N}\left(\boldsymbol{\theta}; \boldsymbol{\mu}, \boldsymbol{\Sigma}\right) ,$$

*implies that*

$$p\left(\boldsymbol{\theta}|\mathbf{x}\right) = \mathcal{N}\left(\boldsymbol{\theta}; \mathbf{C}^{-1}\left(\mathbf{A}^T\mathbf{V}^{-1}\left(\mathbf{x}-\mathbf{b}\right) + \boldsymbol{\Sigma}^{-1}\boldsymbol{\mu}\right), \mathbf{C}^{-1}\right) , \tag{27}$$

$$p\left(\mathbf{x}\right) = \mathcal{N}\left(\mathbf{x}; \mathbf{A}\boldsymbol{\mu} + \mathbf{b}, \mathbf{V} + \mathbf{A}\boldsymbol{\Sigma}\mathbf{A}^T\right) , \tag{28}$$

$$\mathbf{C} = \mathbf{A}^T\mathbf{V}^{-1}\mathbf{A} + \boldsymbol{\Sigma}^{-1} .$$

*Proof.* Let's start by proving (27). First, we note that

$$p\left(\boldsymbol{\theta}|\mathbf{x}\right) = \frac{p\left(\mathbf{x}|\boldsymbol{\theta}\right)\cdot p\left(\boldsymbol{\theta}\right)}{\int p\left(\mathbf{x}|\widetilde{\boldsymbol{\theta}}\right)\cdot p\left(\widetilde{\boldsymbol{\theta}}\right) d\widetilde{\boldsymbol{\theta}}} = \frac{\mathcal{N}\left(\mathbf{x}; \mathbf{A}\boldsymbol{\theta} + \mathbf{b}, \mathbf{V}\right)\cdot \mathcal{N}\left(\boldsymbol{\theta}; \boldsymbol{\mu}, \boldsymbol{\Sigma}\right)}{\int \mathcal{N}\left(\mathbf{x}; \mathbf{A}\widetilde{\boldsymbol{\theta}} + \mathbf{b}, \mathbf{V}\right)\cdot \mathcal{N}\left(\widetilde{\boldsymbol{\theta}}; \boldsymbol{\mu}, \boldsymbol{\Sigma}\right) d\widetilde{\boldsymbol{\theta}}} ,$$

where only the numerator depends on $\boldsymbol{\theta}$. Thus, the numerator is given by

$$c\cdot \exp\left(-\frac{1}{2}\left[\left(\left(\mathbf{x}-\mathbf{b}\right) - \mathbf{A}\boldsymbol{\theta}\right)^{\mathrm{T}}\mathbf{V}^{-1}\left(\left(\mathbf{x}-\mathbf{b}\right) - \mathbf{A}\boldsymbol{\theta}\right) + \left(\boldsymbol{\theta}-\boldsymbol{\mu}\right)^{\mathrm{T}}\boldsymbol{\Sigma}^{-1}\left(\boldsymbol{\theta}-\boldsymbol{\mu}\right)\right]\right) ,$$

where $c = |2\pi\mathbf{V}|^{-\frac{1}{2}}\cdot|2\pi\boldsymbol{\Sigma}|^{-\frac{1}{2}}$ is independent of $\mathbf{x}$ and $\boldsymbol{\theta}$. Using Theorem A.86 in [14], this quadratic form in $\boldsymbol{\theta}$ can be rewritten as

$$c\cdot \exp\left(-\frac{1}{2}\left[\left(\boldsymbol{\theta}-\mathbf{c}\right)^{\mathrm{T}}\mathbf{C}\left(\boldsymbol{\theta}-\mathbf{c}\right) + d\left(\mathbf{x}\right)\right]\right) ,$$

where

$$\mathbf{C} = \mathbf{A}^{\mathrm{T}}\mathbf{V}^{-1}\mathbf{A} + \boldsymbol{\Sigma}^{-1} ,$$

$$\mathbf{C}\mathbf{c} = \mathbf{A}^{\mathrm{T}}\mathbf{V}^{-1}\left(\mathbf{x}-\mathbf{b}\right) + \boldsymbol{\Sigma}^{-1}\boldsymbol{\mu} ,$$

$$d\left(\mathbf{x}\right) = \left(\mathbf{x}-\mathbf{b}-\mathbf{A}\boldsymbol{\mu}\right)^{\mathrm{T}}\left(\mathbf{V} + \mathbf{A}\boldsymbol{\Sigma}\mathbf{A}^{\mathrm{T}}\right)^{-1}\left(\mathbf{x}-\mathbf{b}-\mathbf{A}\boldsymbol{\mu}\right) . \tag{29}$$

Since $d(\mathbf{x})$ does not depend on $\boldsymbol{\theta}$, it get incorporated into the normalization constant which proves (27).

In order to prove (28), note that

$$p\left(\mathbf{x}\right) = \int p\left(\mathbf{x}|\widetilde{\boldsymbol{\theta}}\right)\cdot p\left(\widetilde{\boldsymbol{\theta}}\right) d\widetilde{\boldsymbol{\theta}}$$

$$= \int \mathcal{N}\left(\mathbf{x}; \mathbf{A}\widetilde{\boldsymbol{\theta}} + \mathbf{b}, \mathbf{V}\right)\cdot \mathcal{N}\left(\widetilde{\boldsymbol{\theta}}; \boldsymbol{\mu}, \boldsymbol{\Sigma}\right) d\widetilde{\boldsymbol{\theta}}$$

$$= \int c\cdot \exp\left(-\frac{1}{2}\left[\left(\widetilde{\boldsymbol{\theta}}-\mathbf{c}\right)^{\mathrm{T}}\mathbf{C}\left(\widetilde{\boldsymbol{\theta}}-\mathbf{c}\right) + d\left(\mathbf{x}\right)\right]\right) d\widetilde{\boldsymbol{\theta}}$$

$$= c\cdot \exp\left(-\frac{1}{2}d\left(\mathbf{x}\right)\right)\cdot \int \exp\left(-\frac{1}{2}\left[\left(\widetilde{\boldsymbol{\theta}}-\mathbf{c}\right)^{\mathrm{T}}\mathbf{C}\left(\widetilde{\boldsymbol{\theta}}-\mathbf{c}\right)\right]\right) d\widetilde{\boldsymbol{\theta}}$$

$$= c\cdot \exp\left(-\frac{1}{2}d\left(\mathbf{x}\right)\right)\cdot |2\pi\mathbf{C}^{-1}|^{\frac{1}{2}}$$

$$= \tilde{c}\cdot \exp\left(-\frac{1}{2}\left(\left(\mathbf{x}-\left(\mathbf{A}\boldsymbol{\mu}+\mathbf{b}\right)\right)^{\mathrm{T}}\left(\mathbf{V} + \mathbf{A}\boldsymbol{\Sigma}\mathbf{A}^{\mathrm{T}}\right)^{-1}\left(\mathbf{x}-\left(\mathbf{A}\boldsymbol{\mu}+\mathbf{b}\right)\right)\right)\right)$$

$$= \mathcal{N}\left(\mathbf{x}; \mathbf{A}\boldsymbol{\mu} + \mathbf{b}, \mathbf{V} + \mathbf{A}\boldsymbol{\Sigma}\mathbf{A}^{\mathrm{T}}\right) ,$$

where the penultimate line follows again from Theorem A.86 in [14] with $d(\mathbf{x})$ defined in (29).   □

**Theorem 2** (Woodbury formula). *Let $\mathbf{C}$ be an invertible $n\times n$ matrix. Then, for any $\mathbf{A} \in \mathbb{R}^{n\times k}$ and $\mathbf{B} \in \mathbb{R}^{k\times n}$,*

$$\left(\mathbf{C} + \mathbf{A}\mathbf{B}\right)^{-1} = \mathbf{C}^{-1} - \mathbf{C}^{-1}\mathbf{A}\left(\mathbf{I} + \mathbf{B}\mathbf{C}^{-1}\mathbf{A}\right)^{-1}\mathbf{B}\mathbf{C}^{-1} .$$

## C  Invariance gap and posterior-predictive equivalence

In this section, we show that the posterior predictive distribution of the mean-field and invariance-abiding distribution are identical (for identical parametrisation of the likelihood approximation) as stated in (9), and that the difference in the respective KL regularisation terms can be quantified by the KL defined in (10).

We first discuss the conditions from (7) and (8). The goal is to use the invariance property of the likelihood function (4). Unfortunately, the prior does not generally have the same invariance. Yet, the mean-field approximate posterior defined as the product between prior and likelihood approximation can still have the invariance property, as we show for permutation and translation invariance with mean-field Gaussian likelihood approximation and prior. This condition is defined in (7) for each mean-field posterior in the integral over $\mathbf{r}$:

$$
\begin{aligned}
q_{\mathrm{mix}}(\mathbf{w}; \boldsymbol{\theta}) &= Z_{\mathrm{mix}}^{-1}\, p(\mathbf{w}) \cdot \int p(\mathbf{r})\, g_0(t(\mathbf{w},\mathbf{r}); \boldsymbol{\theta})d\mathbf{r} = \int p(\mathbf{r})\, \frac{1}{Z_{\mathrm{mix}}}\, p(\mathbf{w}) \cdot g_0(t(\mathbf{w},\mathbf{r}); \boldsymbol{\theta})d\mathbf{r} \\
&= \int p(\mathbf{r})\, \frac{1}{Z_{\mathrm{mix}}}\, p(t(\mathbf{w},\varphi(\mathbf{r}))) \cdot g_0(t(\mathbf{w},\varphi(\mathbf{r})); \boldsymbol{\theta})d\mathbf{r} \\
&= \int p(\mathbf{r})\, \frac{Z_0(\mathbf{r})}{Z_{\mathrm{mix}}}\, q_0(t(\mathbf{w},\varphi(\mathbf{r})); \boldsymbol{\theta})d\mathbf{r},
\end{aligned}
$$

where $Z_{\mathrm{mix}}$ and $Z_0(\mathbf{r})$ are normalisation constants. The second line introduced the assumption that there exists a surjective mapping $\mathbf{r}' = \varphi(\mathbf{r})$ such that the product of the untransformed prior $p(\mathbf{w})$ and the transformed likelihood approximation $g_0(t(\mathbf{w},\varphi(\mathbf{r})); \boldsymbol{\theta})$ are identical to the product of the transformed prior and likelihood approximation with $\mathbf{r}'$, i.e. as stated in the main text,

$$
\forall \mathbf{r} \sim p(\mathbf{r}) : p(\mathbf{w}) \cdot g_0(t(\mathbf{w},\mathbf{r}); \boldsymbol{\theta}) = p(t(\mathbf{w},\varphi(\mathbf{r}))) \cdot g_0(t(\mathbf{w},\varphi(\mathbf{r})); \boldsymbol{\theta}).
$$

This condition may seem quite limiting, however, we show that this condition holds for *permutation* invariance with the isotropic Gaussian prior and for *translation* invariance with Gaussian priors and Gaussian likelihood approximation.

**Condition** (7) **and** (8) **for permutation invariance.**  Consider first permutation invariance (cf. App. E). It is easy to see that for the isotropic Gaussian prior: $p(\mathbf{w}) = p(\mathbf{P_r}\mathbf{w})$. Also, note that $\mathcal{G}(\mathbf{P}\mathbf{w}; \boldsymbol{\eta}, \boldsymbol{\Lambda}) = \mathcal{G}(\mathbf{w}; \mathbf{P}^{\mathsf{T}}\boldsymbol{\eta}, \mathbf{P}^{\mathsf{T}}\boldsymbol{\Lambda}\mathbf{P})$. Thus, the product between the untransformed Gaussian prior and the permuted Gaussian likelihood approximation is

$$
p(\mathbf{w}) \cdot g_0(\mathbf{P}\mathbf{w}) = p(\mathbf{P}\mathbf{w}) \cdot g_0(\mathbf{P}\mathbf{w}) = \mathcal{G}(\mathbf{P}\mathbf{w}; \boldsymbol{\eta}_{\mathrm{p}}, \boldsymbol{\Lambda}_{\mathrm{p}}) \cdot \mathcal{G}(\mathbf{P}\mathbf{w}; \boldsymbol{\eta}_g, \boldsymbol{\Lambda}_g) \tag{30}
$$

$$
= \mathcal{G}(\mathbf{w}; \mathbf{P}^{\mathsf{T}}\boldsymbol{\eta}_{\mathrm{p}}, \mathbf{P}^{\mathsf{T}}\boldsymbol{\Lambda}_{\mathrm{p}}\mathbf{P}) \cdot \mathcal{G}(\mathbf{w}; \mathbf{P}^{\mathsf{T}}\boldsymbol{\eta}_g, \mathbf{P}^{\mathsf{T}}\boldsymbol{\Lambda}_g\mathbf{P}) \tag{31}
$$

$$
\propto \mathcal{G}(\mathbf{w}; \mathbf{P}^{\mathsf{T}}(\boldsymbol{\eta}_{\mathrm{p}} + \boldsymbol{\eta}_g), \mathbf{P}^{\mathsf{T}}(\boldsymbol{\Lambda}_{\mathrm{p}} + \boldsymbol{\Lambda}_g)\mathbf{P}) \tag{32}
$$

$$
= \mathcal{G}(\mathbf{P}\mathbf{w}; \boldsymbol{\eta}_{\mathrm{p}} + \boldsymbol{\eta}_g, \boldsymbol{\Lambda}_{\mathrm{p}} + \boldsymbol{\Lambda}_g) = q_0(\mathbf{P}\mathbf{w}). \tag{33}
$$

Hence, for permutation invariance with the isotropic Gaussian prior and Gaussian likelihood approximation, we have

$$
\forall \mathbf{r} \sim p(\mathbf{r}) : p(\mathbf{w}) \cdot g_0(\mathbf{P_r}\mathbf{w}) = p(\mathbf{P_r}\mathbf{w}) \cdot g_0(\mathbf{P_r}\mathbf{w}) \propto q_0(\mathbf{P_r}\mathbf{w}). \tag{34}
$$

The invariance transformation is thus simply $t(\mathbf{w},\varphi(\mathbf{r})) = t(\mathbf{w},\mathbf{r}) = \mathbf{P_r}\mathbf{w}$ (i.e. we do not need the mapping $\varphi$). This is because the prior has no preference over the permutation-induced modes of the likelihood. Thus, we have

$$
\forall \mathbf{r} \sim p(\mathbf{r}) : \left| \det \frac{\partial t(\mathbf{w},\mathbf{r})}{\partial \mathbf{w}} \right|^{-1} = \left| \det \frac{\partial \mathbf{P_r}\mathbf{w}}{\partial \mathbf{w}} \right|^{-1} = |\det \mathbf{P_r}|^{-1} = 1 \,.
$$

**Condition** (7) **and** (8) **for translation invariance.**  Next, consider translation invariance, and note that $\mathcal{N}(\mathbf{w} - \mathbf{v}; \boldsymbol{\mu}, \boldsymbol{\Sigma}) = \mathcal{N}(\mathbf{w}; \boldsymbol{\mu} + \mathbf{v}, \boldsymbol{\Sigma})$, and, consequently, $\mathcal{G}(\mathbf{w} - \mathbf{v}; \boldsymbol{\eta}, \boldsymbol{\Lambda}) = \mathcal{G}(\mathbf{w}; \boldsymbol{\eta} + \boldsymbol{\Lambda}\mathbf{v}, \boldsymbol{\Lambda})$. It is easier to show directly that the product between the untransformed Gaussian prior and translated

Gaussian likelihood approximation can be written as a Gaussian posterior translated, because:

$$p(\mathbf{w}) \cdot g_0(\mathbf{w} - \mathbf{v}) = \mathcal{G}(\mathbf{w}; \boldsymbol{\eta}_{\mathrm{p}}, \boldsymbol{\Lambda}_{\mathrm{p}}) \cdot \mathcal{G}(\mathbf{w} - \mathbf{v}; \boldsymbol{\eta}_g, \boldsymbol{\Lambda}_g) \tag{35}$$

$$= \mathcal{G}(\mathbf{w}; \boldsymbol{\eta}_{\mathrm{p}}, \boldsymbol{\Lambda}_{\mathrm{p}}) \cdot \mathcal{G}(\mathbf{w}; \boldsymbol{\eta}_g + \boldsymbol{\Lambda}_g \mathbf{v}, \boldsymbol{\Lambda}_g) \tag{36}$$

$$\propto \mathcal{G}(\mathbf{w}; \boldsymbol{\eta}_{\mathrm{p}} + \boldsymbol{\eta}_g + \boldsymbol{\Lambda}_g \mathbf{v}, \boldsymbol{\Lambda}_{\mathrm{p}} + \boldsymbol{\Lambda}_g) \tag{37}$$

$$= \mathcal{G}\Big(\mathbf{w}; \boldsymbol{\eta}_{\mathrm{p}} + \boldsymbol{\eta}_g + \overbrace{(\boldsymbol{\Lambda}_{\mathrm{p}} + \boldsymbol{\Lambda}_g)(\boldsymbol{\Lambda}_{\mathrm{p}} + \boldsymbol{\Lambda}_g)^{-1}}^{\mathbf{I}} \boldsymbol{\Lambda}_g \mathbf{v}, \boldsymbol{\Lambda}_{\mathrm{p}} + \boldsymbol{\Lambda}_g\Big) \tag{38}$$

$$= \mathcal{G}\left(\mathbf{w} - (\boldsymbol{\Lambda}_{\mathrm{p}} + \boldsymbol{\Lambda}_g)^{-1} \boldsymbol{\Lambda}_g \mathbf{v}; \boldsymbol{\eta}_{\mathrm{p}} + \boldsymbol{\eta}_g, \boldsymbol{\Lambda}_{\mathrm{p}} + \boldsymbol{\Lambda}_g\right) \tag{39}$$

$$= q_0\left(\mathbf{w} - (\boldsymbol{\Lambda}_{\mathrm{p}} + \boldsymbol{\Lambda}_g)^{-1} \boldsymbol{\Lambda}_g \mathbf{v}\right). \tag{40}$$

Since the precision matrices are diagonal, $((\boldsymbol{\Lambda}_{\mathrm{p}} + \boldsymbol{\Lambda}_g)^{-1} \boldsymbol{\Lambda}_g)\mathbf{Br} = \mathbf{B}((\boldsymbol{\Lambda}_{\mathrm{p}} + \boldsymbol{\Lambda}_g)^{-1} \boldsymbol{\Lambda}_g)\mathbf{r}$. Consequently, for the translation invariance $g_0(\mathbf{w}) = g_0(t(\mathbf{w}, \mathbf{r})) = g_0(\mathbf{w} - \mathbf{Br})$, we have

$$\forall \mathbf{r} \sim p(\mathbf{r}) : p(\mathbf{w}) \cdot g_0(\mathbf{w} - \mathbf{Br}) = p(\mathbf{w} - \mathbf{Br}') \cdot g_0(\mathbf{w} - \mathbf{Br}') \propto q_0(\mathbf{w} - \mathbf{Br}'), \tag{41}$$

where $\mathbf{r}' = \varphi(\mathbf{r}) = \left((\boldsymbol{\Lambda}_{\mathrm{p}} + \boldsymbol{\Lambda}_g)^{-1} \boldsymbol{\Lambda}_g\right) \mathbf{r}$. Thus, we have

$$\forall \mathbf{r} \sim p(\mathbf{r}) : \left|\det \frac{\partial t(\mathbf{w}, \mathbf{r})}{\partial \mathbf{w}}\right|^{-1} = \left|\det \frac{\partial (\mathbf{w} - \mathbf{Br})}{\partial \mathbf{w}}\right|^{-1} = |\det \mathbf{I}|^{-1} = 1.$$

**Lemma 1.** *For any distribution $p(\mathbf{w})$, likelihood approximation $g_0(\mathbf{w}; \boldsymbol{\theta})$, $q_0(\mathbf{w}; \boldsymbol{\theta})$ as defined in* (6), *$q_{\mathrm{mix}}(\mathbf{w}; \boldsymbol{\theta})$ as defined in* (6b) *and $p(\mathbf{r})$, assume there exists a mapping $\varphi : \mathbf{r} \mapsto \mathbf{r}'$ such that*

$$\forall \mathbf{r} \sim p(\mathbf{r}) : p(\mathbf{w}) \cdot g_0(t(\mathbf{w}, \mathbf{r}); \boldsymbol{\theta}) = p(t(\mathbf{w}, \varphi(\mathbf{r}))) \cdot g_0(t(\mathbf{w}, \varphi(\mathbf{r})); \boldsymbol{\theta}), \tag{42}$$

*as well as*

$$\forall \mathbf{r} \sim p(\mathbf{r}) : \left|\det \frac{\partial t(\mathbf{w}, \mathbf{r})}{\partial \mathbf{w}}\right|^{-1} = 1. \tag{43}$$

*Then,*

$$\mathbb{E}_{\mathbf{w} \sim q_{\mathrm{mix}}(\mathbf{w}; \boldsymbol{\theta})} [\ln p(\mathcal{D} \,|\, \mathbf{w})] = \mathbb{E}_{\mathbf{w} \sim q_0(\mathbf{w}; \boldsymbol{\theta})} [\ln p(\mathcal{D} \,|\, \mathbf{w})]. \tag{44}$$

*Proof.* The lemma can be proven by applying the change-of-variables formula and using the invariance property (42) of $g(\cdot; \boldsymbol{\theta})$:

$$\mathbb{E}_{\mathbf{w} \sim q_{\mathrm{mix}}(\mathbf{w}; \boldsymbol{\theta})} [\ln p(\mathcal{D} \,|\, \mathbf{w})] \tag{45}$$

$$= \int \overbrace{\int p(\mathbf{r}) \frac{Z_0(\mathbf{r})}{Z_{\mathrm{mix}}} q_0(t(\mathbf{w}, \varphi(\mathbf{r})); \boldsymbol{\theta}) \, d\mathbf{r}}^{q_{\mathrm{mix}}(\mathbf{w})} \ln [p(\mathcal{D} \,|\, \mathbf{w})] \, d\mathbf{w} \tag{46}$$

$$= \int p(\mathbf{r}) \frac{Z_0(\mathbf{r})}{Z_{\mathrm{mix}}} \int q_0(t(\mathbf{w}, \varphi(\mathbf{r})); \boldsymbol{\theta}) \ln [p(\mathcal{D} \,|\, \mathbf{w})] \, d\mathbf{w} \, d\mathbf{r} \tag{47}$$

$$= \int p(\mathbf{r}) \frac{Z_0(\mathbf{r})}{Z_{\mathrm{mix}}} \int q_0(t(\mathbf{w}, \varphi(\mathbf{r})); \boldsymbol{\theta}) \ln [p(\mathcal{D} \,|\, t(\mathbf{w}, \varphi(\mathbf{r})))] \, d\mathbf{w} \, d\mathbf{r} \tag{48}$$

$$= \int p(\mathbf{r}) \frac{Z_0(\mathbf{r})}{Z_{\mathrm{mix}}} \int q_0(\mathbf{w}; \boldsymbol{\theta}) \overbrace{\left|\det \frac{\partial t(\mathbf{w}, \varphi(\mathbf{r}))}{\partial \mathbf{w}}\right|^{-1}}^{1} \ln [p(\mathcal{D} \,|\, \mathbf{w})] \, d\mathbf{w} \, d\mathbf{r} \tag{49}$$

$$= \overbrace{\int p(\mathbf{r}) \frac{Z_0(\mathbf{r})}{Z_{\mathrm{mix}}} d\mathbf{r}}^{1} \int q_0(\mathbf{w}; \boldsymbol{\theta}) \ln [p(\mathcal{D} \,|\, \mathbf{w})] \, d\mathbf{w} \tag{50}$$

$$= \mathbb{E}_{\mathbf{w} \sim q_0} [\ln p(\mathcal{D} \,|\, \mathbf{w})]. \tag{51}$$

where the second line uses (42), the third line changes the order of integration (Fubini's theorem), the fourth line uses the invariance property $\ln p(\mathcal{D} \,|\, \mathbf{w}) = \ln p(\mathcal{D} \,|\, t(\mathbf{w}, \varphi(\mathbf{r})))$, the fifth line then applies

the change of variables theorem together with (43), the sixth line then uses again the invariance property of the log likelihood, and the seventh line notes that the integral over the normalisation constants $Z_0(\mathbf{r})$ cancels with the normalisation constant $Z_{\mathrm{mix}}$ of the invariance-abiding posterior, since

$$\int p(\mathbf{r}) Z_0(\mathbf{r}) d\mathbf{r} = \int \int p(\mathbf{r}) \, p(\mathbf{w}) \cdot g_0(t(\mathbf{w}, \mathbf{r})) \, d\mathbf{w} d\mathbf{r} \tag{52}$$

$$= \int p(\mathbf{w}) \cdot \int p(\mathbf{r}) \, g_0(t(\mathbf{w}, \mathbf{r})) \, d\mathbf{r} d\mathbf{w} \tag{53}$$

$$= \int p(\mathbf{w}) \cdot g_{\mathrm{mix}}(\mathbf{w}) \, d\mathbf{w} = Z_{\mathrm{mix}}, \tag{54}$$

where we changed the integration order, resulting in the normalisation constant of the invariance-abiding likelihood approximation. $\qquad\square$

**Lemma 2.** *For any distribution $p(\mathbf{w})$, likelihood approximation $g_0(\mathbf{w}; \boldsymbol{\theta})$, $q_0(\mathbf{w}; \boldsymbol{\theta})$ as defined in (6), $q_{\mathrm{mix}}(\mathbf{w}; \boldsymbol{\theta})$ as defined in (6b) and $p(\mathbf{r})$, assume there exist two mappings $t : \mathbf{w} \times \mathbf{r} \mapsto \mathbf{w}'$ and $\varphi : \mathbf{r} \mapsto \mathbf{r}'$ such that (42) and (43) hold. Then,*

$$\mathrm{KL}\left[q_0 \,||\, p\right] - \mathrm{KL}\left[q_{\mathrm{mix}} \,||\, p\right] = \mathrm{KL}\left[q_0 \,||\, q_{\mathrm{mix}}\right] . \tag{55}$$

*Proof.* First, note that

$$\mathrm{KL}\left[q_0 \,||\, p\right] = \mathbb{E}_{\mathbf{w} \sim q_0}\left[\ln\left[Z_0 \, g_0(\mathbf{w})\right]\right] , \tag{56}$$

$$\mathrm{KL}\left[q_{\mathrm{mix}} \,||\, p\right] = \mathbb{E}_{\mathbf{w} \sim q_{\mathrm{mix}}}\left[\ln\left[Z_{\mathrm{mix}} \, g_{\mathrm{mix}}(\mathbf{w})\right]\right] . \tag{57}$$

Now, in the latter KL, we expand the distribution $q_{\mathrm{mix}}(\mathbf{w})$ as in (42),

$$\mathrm{KL}\left[q_{\mathrm{mix}} \,||\, p\right] = \int \overbrace{\int p(\mathbf{r}) \frac{Z_0(\mathbf{r})}{Z_{\mathrm{mix}}} q_0(t(\mathbf{w}, \varphi(\mathbf{r})); \boldsymbol{\theta}) d\mathbf{r}}^{q_{\mathrm{mix}}(\mathbf{w})} \ln\left[Z_{\mathrm{mix}} \, g_{\mathrm{mix}}(\mathbf{w})\right] \, d\mathbf{w} \tag{58}$$

$$= \int \int p(\mathbf{r}) \frac{Z_0(\mathbf{r})}{Z_{\mathrm{mix}}} q_0(t(\mathbf{w}, \varphi(\mathbf{r})); \boldsymbol{\theta}) \ln\left[Z_{\mathrm{mix}} \, g_{\mathrm{mix}}(\mathbf{w})\right] \, d\mathbf{r} d\mathbf{w} \tag{59}$$

$$= \int \int p(\mathbf{r}) \frac{Z_0(\mathbf{r})}{Z_{\mathrm{mix}}} q_0(t(\mathbf{w}, \varphi(\mathbf{r})); \boldsymbol{\theta}) \ln\left[Z_{\mathrm{mix}} \, g_{\mathrm{mix}}(t(\mathbf{w}, \varphi(\mathbf{r})))\right] \, d\mathbf{r} d\mathbf{w} \tag{60}$$

$$= \int \int p(\mathbf{r}) \frac{Z_0(\mathbf{r})}{Z_{\mathrm{mix}}} q_0(\mathbf{w}; \boldsymbol{\theta}) \overbrace{\left|\det \frac{\partial t(\mathbf{w}, \varphi(\mathbf{r}))}{\partial \mathbf{w}}\right|^{-1}}^{1} \ln\left[Z_{\mathrm{mix}} \, g_{\mathrm{mix}}(\mathbf{w})\right] \, d\mathbf{r} d\mathbf{w} \tag{61}$$

$$= \int q_0(\mathbf{w}; \boldsymbol{\theta}) \overbrace{\int p(\mathbf{r}) \frac{Z_0(\mathbf{r})}{Z_{\mathrm{mix}}} \, d\mathbf{r}}^{1} \ln\left[Z_{\mathrm{mix}} \, g_{\mathrm{mix}}(\mathbf{w})\right] d\mathbf{w} \tag{62}$$

$$= \int q_0(\mathbf{w}; \boldsymbol{\theta}) \ln\left[Z_{\mathrm{mix}} \, g_{\mathrm{mix}}(\mathbf{w})\right] \, d\mathbf{w}' , \tag{63}$$

where the third line uses the invariance property of the invariance-abiding likelihood approximation, $\forall \mathbf{r} : g_{\mathrm{mix}}(t(\mathbf{w}, \varphi(\mathbf{r}))) = g_{\mathrm{mix}}(\mathbf{w})$, the change of variables formula is applied to the fourth line for the volume-preserving transformation, and the fifth line re-arranges the integration, noting that the integral over normalisation constants equals one.

The proposition follows by taking the difference between the regularisation terms corresponding to the respective ELBO objectives:

$$\mathrm{KL}\left[q_0 \,||\, p\right] - \mathrm{KL}\left[q_{\mathrm{mix}} \,||\, p\right] = \mathbb{E}_{\mathbf{w} \sim q_0}\left[\ln \frac{Z_0 \, g_0(\mathbf{w})}{Z_{\mathrm{mix}} g_{\mathrm{mix}}(\mathbf{w})}\right] \tag{64}$$

$$= \mathbb{E}_{\mathbf{w} \sim q_0}\left[\ln \frac{Z_0 \, p(\mathbf{w}) g_0(\mathbf{w})}{Z_{\mathrm{mix}} p(\mathbf{w}) g_{\mathrm{mix}}(\mathbf{w})}\right] = \mathrm{KL}\left[q_0 \,||\, q_{\mathrm{mix}}\right] . \tag{65}$$

$$\square$$

# D   Translation invariance in linear models

In this section, we derive the results from Sec. 4 for a Bayesian linear regression with a single input vector $\mathbf{x}$ and corresponding target observation $y$. The result stated in Sec. 4 for $\mathbf{x} = \mathbf{1}$ follows as a special case.

We assume we have $K$ latent variables $\mathbf{w} = [w_1, \ldots, w_K]^\mathsf{T}$ and one observation $\mathbf{x} = [x_1, \ldots, x_K]^\mathsf{T}$. We further assume that the likelihood $p(y, |\mathbf{w}, \mathbf{x})$ only depend on their inner product, that is, $\mathbf{x}^\mathsf{T}\mathbf{w}$. Then we know that any change in $\mathbf{w}$, which leaves the sum of the elements weighted by $\mathbf{x}$ unaffected, does not change the likelihood. For example $\mathbf{w}' = [w_1 + \Delta, w_2 - \Delta \cdot \frac{x_1}{x_2}, w_3, \ldots, w_K]^\mathsf{T}$ has the exact same likelihood than $\mathbf{w} = [w_1, w_2, w_3, \ldots, w_K]^\mathsf{T}$. In general, we can model this translation invariance using an $K - 1$ dimensional vector $\boldsymbol{\Delta} \in \mathbb{R}^{K-1}$ and noting that

$$
\mathbf{x}^\mathsf{T}\mathbf{w} = \mathbf{x}^\mathsf{T}\left(\mathbf{w} + \underbrace{\begin{bmatrix} \mathbf{I} \\ -x_K^{-1}\mathbf{x}_{K-1}^\mathsf{T} \end{bmatrix}}_{\mathbf{B}} \boldsymbol{\Delta}\right),
$$

where we used $\mathbf{x}_{K-1} := [x_1, \ldots, x_{K-1}]^\mathsf{T}$ because

$$
\mathbf{x}^\mathsf{T}\mathbf{B}\boldsymbol{\Delta} = \begin{bmatrix} \mathbf{x}_{K-1}^\mathsf{T} & x_K \end{bmatrix}\begin{bmatrix} \boldsymbol{\Delta} \\ -x_K^{-1}\mathbf{x}_{K-1}^\mathsf{T}\boldsymbol{\Delta} \end{bmatrix} = 0.
$$

## D.1   Likelihood Model

Now let us assume that we approximate the likelihood by a function that has Gaussian shape, that is $p(y, |\mathbf{w}, \mathbf{x}) \approx q(\mathbf{w}) \propto \mathcal{N}(\mathbf{w}; \mathbf{m}, \mathbf{V})$. In order to model that the likelihood is translation invariant, we compute the marginal $q(\mathbf{w} - \mathbf{B}\boldsymbol{\Delta})$ over $p(\boldsymbol{\Delta}) = \mathcal{N}(\boldsymbol{\Delta}; \mathbf{0}, \beta^2\mathbf{I})$ and considering the case of $\beta \to \infty$, that is

$$
q_\beta(\mathbf{w}) := \int q(\mathbf{w} - \mathbf{B}\boldsymbol{\Delta}) \cdot p(\boldsymbol{\Delta})\, d\boldsymbol{\Delta}
$$
$$
= \int \mathcal{N}(\mathbf{w}; \mathbf{m} + \mathbf{B}\boldsymbol{\Delta}, \mathbf{V}) \cdot \mathcal{N}(\boldsymbol{\Delta}; \mathbf{0}, \beta^2\mathbf{I})\, d\boldsymbol{\Delta}.
$$

According to Theorem 1, for any $\beta \in \mathbb{R}^+$ this is another Gaussian given by

$$
q_\beta(\mathbf{w}) = \mathcal{N}\left(\mathbf{w}; \mathbf{m} + \mathbf{B}\mathbf{0}, \underbrace{\mathbf{V} + \beta\mathbf{B} \cdot \beta\mathbf{B}^\mathsf{T}}_{\mathbf{V}_\beta}\right)
$$
$$
= \mathcal{N}\left(\mathbf{w}; \mathbf{m}, \mathbf{V} + \beta^2 \cdot \begin{bmatrix} \mathbf{I} & -x_K^{-1}\mathbf{x}_{K-1} \\ -x_K^{-1}\mathbf{x}_{K-1}^\mathsf{T} & x_K^{-2}\mathbf{x}_{K-1}^\mathsf{T}\mathbf{x}_{K-1} \end{bmatrix}\right).
$$

Note that $\lim_{\beta \to \infty} q_\beta(\mathbf{w}) = g_{\mathrm{mix}}(\mathbf{w})$ as defined in Subsection 4.1. Using the Woodbury formula in Theorem 2, the inverse of the covariance can be re-written as

$$
\mathbf{V}_\beta^{-1} := \left(\mathbf{V} + \beta\mathbf{B} \cdot \beta\mathbf{B}^\mathsf{T}\right)^{-1}
$$
$$
= \mathbf{V}^{-1} - \beta^2 \cdot \mathbf{V}^{-1}\mathbf{B}\left(\mathbf{I} + \beta^2\mathbf{B}^\mathsf{T}\mathbf{V}^{-1}\mathbf{B}\right)^{-1}\mathbf{B}^\mathsf{T}\mathbf{V}^{-1}
$$
$$
= \mathbf{V}^{-1} - \beta^2 \cdot \mathbf{V}^{-1}\mathbf{B}\left(\beta^2\left(\beta^{-2}\mathbf{I} + \mathbf{B}^\mathsf{T}\mathbf{V}^{-1}\mathbf{B}\right)\right)^{-1}\mathbf{B}^\mathsf{T}\mathbf{V}^{-1}
$$
$$
= \mathbf{V}^{-1} - \mathbf{V}^{-1}\mathbf{B}\left(\beta^{-2}\mathbf{I} + \mathbf{B}^\mathsf{T}\mathbf{V}^{-1}\mathbf{B}\right)^{-1}\mathbf{B}^\mathsf{T}\mathbf{V}^{-1}.
$$

### D.2 Posterior Model

If we assume a prior $p(\mathbf{w}) = \mathcal{N}(\mathbf{w}; \boldsymbol{\mu}, \boldsymbol{\Sigma})$, then the posterior for the Gaussian approximation is given by

$$
\begin{aligned}
p(\mathbf{w}|\mathbf{x}) &\propto q(\mathbf{w}) \cdot p(\mathbf{w}) \\
&\propto \mathcal{G}\left(\mathbf{w}; \mathbf{V}^{-1}\mathbf{m}, \mathbf{V}^{-1}\right) \cdot \mathcal{G}\left(\mathbf{w}; \boldsymbol{\Sigma}^{-1}\boldsymbol{\mu}, \boldsymbol{\Sigma}^{-1}\right) \\
&= \mathcal{G}\left(\mathbf{w}; \mathbf{V}^{-1}\mathbf{m} + \boldsymbol{\Sigma}^{-1}\boldsymbol{\mu}, \mathbf{V}^{-1} + \boldsymbol{\Sigma}^{-1}\right) \\
&= \mathcal{N}\left(\mathbf{w}; \boldsymbol{\Sigma}(\mathbf{V}+\boldsymbol{\Sigma})^{-1}\mathbf{m} + \mathbf{V}(\mathbf{V}+\boldsymbol{\Sigma})^{-1}\boldsymbol{\mu}, \mathbf{V}(\mathbf{V}+\boldsymbol{\Sigma})^{-1}\boldsymbol{\Sigma}\right),
\end{aligned}
$$

where we used the identity $\left(\boldsymbol{\Sigma}^{-1} + \mathbf{V}^{-1}\right)^{-1} = \mathbf{V}(\mathbf{V}+\boldsymbol{\Sigma})^{-1}\boldsymbol{\Sigma} = \boldsymbol{\Sigma}(\mathbf{V}+\boldsymbol{\Sigma})^{-1}\mathbf{V}$ in the last line. Similarly, if we use the likelihood approximation which has incorporated the translation invariance, we get

$$
\begin{aligned}
p_\beta(\mathbf{w}|\mathbf{x}) &\propto q_\beta(\mathbf{w}) \cdot p(\mathbf{w}) \\
&\propto \mathcal{G}\left(\mathbf{w}; \mathbf{V}_\beta^{-1}\mathbf{m}, \mathbf{V}_\beta^{-1}\right) \cdot \mathcal{G}\left(\mathbf{w}; \boldsymbol{\Sigma}^{-1}\boldsymbol{\mu}, \boldsymbol{\Sigma}^{-1}\right) \\
&= \mathcal{G}\left(\mathbf{w}; \mathbf{V}_\beta^{-1}\mathbf{m} + \boldsymbol{\Sigma}^{-1}\boldsymbol{\mu}, \mathbf{V}_\beta^{-1} + \boldsymbol{\Sigma}^{-1}\right) \\
&= \mathcal{N}\left(\mathbf{w}; \left(\mathbf{V}_\beta^{-1} + \boldsymbol{\Sigma}^{-1}\right)^{-1}\left(\mathbf{V}_\beta^{-1}\mathbf{m} + \boldsymbol{\Sigma}^{-1}\boldsymbol{\mu}\right), \left(\mathbf{V}_\beta^{-1} + \boldsymbol{\Sigma}^{-1}\right)^{-1}\right)
\end{aligned}
$$

Observing that $\lim_{\beta \to \infty} \beta^{-2}\mathbf{I} = \mathbf{0}$ we thus see that

$$
p_\infty(\mathbf{w}|\mathbf{x}) = \mathcal{N}\left(\mathbf{w}; \left(\mathbf{V}_{\mathbf{B}}^{-1} + \boldsymbol{\Sigma}^{-1}\right)^{-1}\left(\mathbf{V}_{\mathbf{B}}^{-1}\mathbf{m} + \boldsymbol{\Sigma}^{-1}\boldsymbol{\mu}\right), \left(\mathbf{V}_{\mathbf{B}}^{-1} + \boldsymbol{\Sigma}^{-1}\right)^{-1}\right), \tag{66}
$$

$$
\mathbf{V}_{\mathbf{B}}^{-1} := \mathbf{V}^{-1} - \mathbf{V}^{-1}\mathbf{B}\left(\mathbf{B}^{\mathsf{T}}\mathbf{V}^{-1}\mathbf{B}\right)^{-1}\mathbf{B}^{\mathsf{T}}\mathbf{V}^{-1}. \tag{67}
$$

Note again that $p_\infty(\mathbf{w}) = q_{\text{mix}}(\mathbf{w}; \boldsymbol{\theta})$ as defined in (15).

### D.2.1 Special Case of Diagonal Likelihood Covariance

Now let us consider the special case where the covariance matrix of the Gaussian likelihood approximation is a diagonal matrix, that is $\mathbf{V} = \text{Diag}(\boldsymbol{\lambda})$. Then, observe that

$$
\mathbf{V}^{-1}\mathbf{B} = \begin{bmatrix} \mathbf{V}_{K-1}^{-1} & \mathbf{0} \\ \mathbf{0} & \lambda_K^{-1} \end{bmatrix} \begin{bmatrix} \mathbf{I} \\ -x_K^{-1}\mathbf{x}_{K-1}^{\mathsf{T}} \end{bmatrix} = \begin{bmatrix} \mathbf{V}_{K-1}^{-1} \\ -\lambda_K^{-1}x_K^{-1}\mathbf{x}_{K-1}^{\mathsf{T}} \end{bmatrix}, \tag{68}
$$

$$
\mathbf{V}^{-1}\mathbf{B}\mathbf{V}_{K-1}\mathbf{x}_{K-1} = \left(\mathbf{V}^{-1}\mathbf{B}\right) \cdot \mathbf{V}_{K-1}\mathbf{x}_{K-1} = \begin{bmatrix} \mathbf{x}_{K-1} \\ -\lambda_K^{-1}x_K^{-1}\mathbf{x}_{K-1}^{\mathsf{T}}\mathbf{V}_{K-1}\mathbf{x}_{K-1} \end{bmatrix}, \tag{69}
$$

$$
\mathbf{B}^{\mathsf{T}}\mathbf{V}^{-1}\mathbf{B} = \begin{bmatrix} \mathbf{I} & -x_K^{-1}\mathbf{x}_{K-1} \end{bmatrix} \begin{bmatrix} \mathbf{V}_{K-1}^{-1} \\ -\lambda_K^{-1}x_K^{-1}\mathbf{x}_{K-1}^{\mathsf{T}} \end{bmatrix} = \mathbf{V}_{K-1}^{-1} + \lambda_K^{-1}x_K^{-2}\mathbf{x}_{K-1}\mathbf{x}_{K-1}^{\mathsf{T}},
$$

where we used the notation $\mathbf{V}_{K-1}^{-1}$ to denote the diagonal $(K-1) \times (K-1)$ matrix with all

$$
\boldsymbol{\lambda}_{K-1} := \left[\lambda_1^{-1}, \lambda_2^{-1}, \ldots, \lambda_{K-1}^{-1}\right]^{\mathsf{T}}
$$

on the diagonal. Thus, using Theorem 2 with $\mathbf{C} = \mathbf{V}_{K-1}^{-1}$, $\mathbf{A} = x_K^{-2}\lambda_K^{-1}\mathbf{x}_{K-1}$ and $\mathbf{B} = \mathbf{x}_{K-1}^{\mathsf{T}}$, we see that

$$
\begin{aligned}
\left(\mathbf{B}^{\mathsf{T}}\mathbf{V}^{-1}\mathbf{B}\right)^{-1} &= \mathbf{V}_{K-1} - \frac{1}{\lambda_K x_K^2 + \mathbf{x}_{K-1}^{\mathsf{T}}\mathbf{V}_{K-1}\mathbf{x}_{K-1}}\mathbf{V}_{K-1}\mathbf{x}_{K-1}\mathbf{x}_{K-1}^{\mathsf{T}}\mathbf{V}_{K-1} \\
&= \mathbf{V}_{K-1} - \frac{1}{\mathbf{x}^{\mathsf{T}}\mathbf{V}\mathbf{x}}\mathbf{V}_{K-1}\mathbf{x}_{K-1}\mathbf{x}_{K-1}^{\mathsf{T}}\mathbf{V}_{K-1},
\end{aligned}
$$

**Covariance $\left(\mathbf{V}_{\mathbf{B}}^{-1} + \boldsymbol{\Sigma}^{-1}\right)$.** Thus, for the covariance (67) of the posterior we have

$$\mathbf{V}^{-1} - \mathbf{V}^{-1}\mathbf{B}\left(\mathbf{B}^{\mathsf{T}}\mathbf{V}^{-1}\mathbf{B}\right)^{-1}\mathbf{B}^{\mathsf{T}}\mathbf{V}^{-1}$$

$$= \mathbf{V}^{-1} - \mathbf{V}^{-1}\mathbf{B}\left[\mathbf{V}_{K-1} - \frac{1}{\mathbf{x}^{\mathsf{T}}\mathbf{V}\mathbf{x}}\mathbf{V}_{K-1}\mathbf{x}_{K-1}\mathbf{x}_{K-1}^{\mathsf{T}}\mathbf{V}_{K-1}\right]\mathbf{B}^{\mathsf{T}}\mathbf{V}^{-1}$$

$$= \mathbf{V}^{-1} - \underbrace{\mathbf{V}^{-1}\mathbf{B}\mathbf{V}_{K-1}\mathbf{B}^{\mathsf{T}}\mathbf{V}^{-1}}_{S} + \underbrace{\frac{1}{\mathbf{x}^{\mathsf{T}}\mathbf{V}\mathbf{x}}\mathbf{V}^{-1}\mathbf{B}\mathbf{V}_{K-1}\mathbf{x}_{K-1}\mathbf{x}_{K-1}^{\mathsf{T}}\mathbf{V}_{K-1}\mathbf{B}^{\mathsf{T}}\mathbf{V}^{-1}}_{T}.$$

Let's focus on the expression $S$ first. Using (68) we have

$$S = \begin{bmatrix} \mathbf{V}_{K-1}^{-1} \\ -\lambda_K^{-1}x_K^{-1}\mathbf{x}_{K-1}^{\mathsf{T}} \end{bmatrix}\mathbf{V}_{K-1}\begin{bmatrix} \mathbf{V}_{K-1}^{-1} & -\lambda_K^{-1}x_K^{-1}\mathbf{x}_{K-1} \end{bmatrix}$$

$$= \begin{bmatrix} \mathbf{V}_{K-1}^{-1} & -\lambda_K^{-1}x_K^{-1}\mathbf{x}_{K-1} \\ -\lambda_K^{-1}x_K^{-1}\mathbf{x}_{K-1}^{\mathsf{T}} & \lambda_K^{-2}x_K^{-2}\mathbf{x}_{K-1}^{\mathsf{T}}\mathbf{V}_{K-1}\mathbf{x}_{K-1} \end{bmatrix}. \tag{70}$$

Similarly, for the expression $T$ using (69) we have

$$T = \frac{1}{\mathbf{x}^{\mathsf{T}}\mathbf{V}\mathbf{x}}\begin{bmatrix} \mathbf{x}_{K-1} \\ -\lambda_K^{-1}x_K^{-1}\mathbf{x}_{K-1}^{\mathsf{T}}\mathbf{V}_{K-1}\mathbf{x}_{K-1} \end{bmatrix}\begin{bmatrix} \mathbf{x}_{K-1} & -\lambda_K^{-1}x_K^{-1}\mathbf{x}_{K-1}^{\mathsf{T}}\mathbf{V}_{K-1}\mathbf{x}_{K-1} \end{bmatrix}$$

$$= \frac{1}{\mathbf{x}^{\mathsf{T}}\mathbf{V}\mathbf{x}}\begin{bmatrix} \mathbf{x}_{K-1}\mathbf{x}_{K-1}^{\mathsf{T}} & -\frac{\mathbf{x}_{K-1}^{\mathsf{T}}\mathbf{V}_{K-1}\mathbf{x}_{K-1}}{\lambda_K x_K}\mathbf{x}_{K-1} \\ -\frac{\mathbf{x}_{K-1}^{\mathsf{T}}\mathbf{V}_{K-1}\mathbf{x}_{K-1}}{\lambda_K x_K}\mathbf{x}_{K-1}^{\mathsf{T}} & \frac{\left(\mathbf{x}_{K-1}^{\mathsf{T}}\mathbf{V}_{K-1}\mathbf{x}_{K-1}\right)^2}{\lambda_K^2 x_K^2} \end{bmatrix}. \tag{71}$$

Putting (70) and (71) together, we get

$$\mathbf{V}_{\mathbf{B}}^{-1} = \frac{1}{\mathbf{x}^{\mathsf{T}}\mathbf{V}\mathbf{x}}\begin{bmatrix} \mathbf{x}_{K-1}\mathbf{x}_{K-1}^{\mathsf{T}} & \left(\frac{\mathbf{x}^{\mathsf{T}}\mathbf{V}\mathbf{x}}{\lambda_K x_K} - \frac{\mathbf{x}_{K-1}^{\mathsf{T}}\mathbf{V}_{K-1}\mathbf{x}_{K-1}}{\lambda_K x_K}\right)\mathbf{x}_{K-1} \\ \left(\frac{\mathbf{x}^{\mathsf{T}}\mathbf{V}\mathbf{x}}{\lambda_K x_K} - \frac{\mathbf{x}_{K-1}^{\mathsf{T}}\mathbf{V}_{K-1}\mathbf{x}_{K-1}}{\lambda_K x_K}\right)\mathbf{x}_{K-1}^{\mathsf{T}} & \frac{\mathbf{x}^{\mathsf{T}}\mathbf{V}\mathbf{x}}{\lambda_K} - \frac{\mathbf{x}^{\mathsf{T}}\mathbf{V}\mathbf{x}\cdot\mathbf{x}_{K-1}^{\mathsf{T}}\mathbf{V}_{K-1}\mathbf{x}_{K-1}}{\lambda_K^2 x_K^2} + \frac{\left(\mathbf{x}_{K-1}^{\mathsf{T}}\mathbf{V}_{K-1}\mathbf{x}_{K-1}\right)^2}{\lambda_K^2 x_K^2} \end{bmatrix}$$

$$= \frac{1}{\mathbf{x}^{\mathsf{T}}\mathbf{V}\mathbf{x}}\begin{bmatrix} \mathbf{x}_{K-1}\mathbf{x}_{K-1}^{\mathsf{T}} & x_K\mathbf{x}_{K-1} \\ x_K\mathbf{x}_{K-1}^{\mathsf{T}} & x_K^2 \end{bmatrix}$$

$$= \frac{1}{\mathbf{x}^{\mathsf{T}}\mathbf{V}\mathbf{x}}\mathbf{x}\mathbf{x}^{\mathsf{T}}, \tag{72}$$

where we repeatedly used that $\mathbf{x}^{\mathsf{T}}\mathbf{V}\mathbf{x} - \mathbf{x}_{K-1}^{\mathsf{T}}\mathbf{V}_{K-1}\mathbf{x}_{K-1} = \lambda_K x_K^2$. Thus, the covariance in (66) can be written as

$$\left(\mathbf{V}_{\mathbf{B}}^{-1} + \mathbf{\Sigma}^{-1}\right)^{-1} = \left(\mathbf{\Sigma}^{-1} + \frac{1}{\mathbf{x}^{\mathsf{T}}\mathbf{V}\mathbf{x}}\mathbf{x}\mathbf{x}^{\mathsf{T}}\right)^{-1}$$

$$= \mathbf{\Sigma} - \frac{1}{\mathbf{x}^{\mathsf{T}}\mathbf{V}\mathbf{x}}\cdot\frac{1}{1 + (\mathbf{x}^{\mathsf{T}}\mathbf{V}\mathbf{x})^{-1}\mathbf{x}^{\mathsf{T}}\mathbf{\Sigma}\mathbf{x}}\cdot\mathbf{\Sigma}\mathbf{x}\mathbf{x}^{\mathsf{T}}\mathbf{\Sigma}$$

$$= \mathbf{\Sigma} - \frac{1}{\mathbf{x}^{\mathsf{T}}\left(\mathbf{V}+\mathbf{\Sigma}\right)\mathbf{x}}\cdot\left(\mathbf{\Sigma}\mathbf{x}\right)\left(\mathbf{\Sigma}\mathbf{x}\right)^{\mathsf{T}}, \tag{73}$$

where we used Theorem 2 in the third step. Note that (14) is a special case of (72) when using $\mathbf{x} = \mathbf{1}$ and observing that $\mathbf{V}\mathbf{1} = \boldsymbol{\lambda}$. Similarly, the covariance in (15) is a special case of (73) when further noticing that $\mathbf{\Sigma}\mathbf{1} = \boldsymbol{\sigma}^2$.

**Mean $\left(\mathbf{V}_{\mathbf{B}}^{-1} + \mathbf{\Sigma}^{-1}\right)^{-1}\left(\mathbf{V}_{\mathbf{B}}^{-1}\mathbf{m} + \mathbf{\Sigma}^{-1}\boldsymbol{\mu}\right)$.** In order to derive an efficient update for the mean of the posterior, please note that by virtue of (72), $\mathbf{V}_{\mathbf{B}}^{-1}$ can be written as $\mathbf{d}\mathbf{d}^{\mathsf{T}}$ with $\mathbf{d} = \left(\mathbf{x}^{\mathsf{T}}\mathbf{V}\mathbf{x}\right)^{-\frac{1}{2}}\cdot\mathbf{x}$.

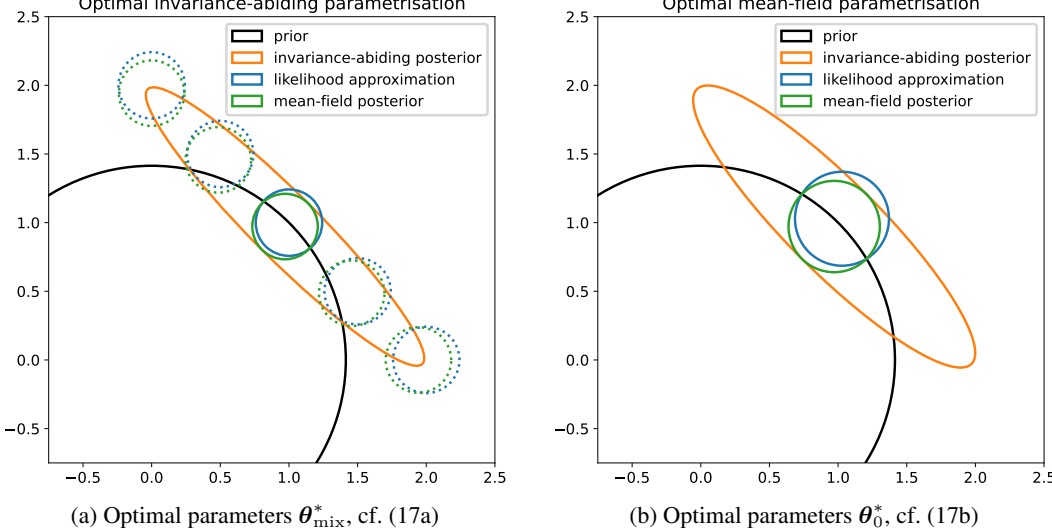

(a) Optimal parameters $\boldsymbol{\theta}^*_{\mathrm{mix}}$, cf. (17a)    (b) Optimal parameters $\boldsymbol{\theta}^*_0$, cf. (17b)

Figure 3: Gaussian likelihood and posterior approximations $q_0$ and $q_{\mathrm{mix}}$ (see (15)) with different parameter optima (cf. (17)). The dotted circles show alternative parameter values that induce the same predictive distribution but do not correspond to one of the optima in (17).

Thus, using (73) we have

$$\left(\mathbf{V_B}^{-1} + \mathbf{\Sigma}^{-1}\right)^{-1}\left(\mathbf{V_B}^{-1}\mathbf{m} + \mathbf{\Sigma}^{-1}\boldsymbol{\mu}\right)$$

$$= \left(\mathbf{\Sigma} - \frac{1}{\mathbf{x}^{\mathrm{T}}\left(\mathbf{V} + \mathbf{\Sigma}\right)\mathbf{x}} \cdot \left(\mathbf{\Sigma}\mathbf{x}\right)\left(\mathbf{\Sigma}\mathbf{x}\right)^{\mathrm{T}}\right)\left(\mathbf{dd}^{\mathrm{T}}\mathbf{m} + \mathbf{\Sigma}^{-1}\boldsymbol{\mu}\right)$$

$$= \mathbf{\Sigma}\mathbf{dd}^{\mathrm{T}}\mathbf{m} + \boldsymbol{\mu} - \frac{1}{\mathbf{x}^{\mathrm{T}}\left(\mathbf{V} + \mathbf{\Sigma}\right)\mathbf{x}} \cdot \left(\mathbf{\Sigma}\mathbf{x}\right)\left(\mathbf{\Sigma}\mathbf{x}\right)^{\mathrm{T}}\mathbf{dd}^{\mathrm{T}}\mathbf{m} - \frac{1}{\mathbf{x}^{\mathrm{T}}\left(\mathbf{V} + \mathbf{\Sigma}\right)\mathbf{x}} \cdot \left(\mathbf{\Sigma}\mathbf{x}\right)\left(\mathbf{\Sigma}\mathbf{x}\right)^{\mathrm{T}}\mathbf{\Sigma}^{-1}\boldsymbol{\mu}$$

$$= \boldsymbol{\mu} + \left(\frac{\mathbf{x}^{\mathrm{T}}\mathbf{m}}{\mathbf{x}^{\mathrm{T}}\mathbf{V}\mathbf{x}}\right) \cdot \left(\mathbf{\Sigma}\mathbf{x}\right) - \frac{\left(\mathbf{\Sigma}\mathbf{x}\right)^{\mathrm{T}}\mathbf{dd}^{\mathrm{T}}\mathbf{m}}{\mathbf{x}^{\mathrm{T}}\left(\mathbf{V} + \mathbf{\Sigma}\right)\mathbf{x}} \cdot \left(\mathbf{\Sigma}\mathbf{x}\right) - \frac{\left(\mathbf{\Sigma}\mathbf{x}\right)^{\mathrm{T}}\mathbf{\Sigma}^{-1}\boldsymbol{\mu}}{\mathbf{x}^{\mathrm{T}}\left(\mathbf{V} + \mathbf{\Sigma}\right)\mathbf{x}} \cdot \left(\mathbf{\Sigma}\mathbf{x}\right)$$

$$= \boldsymbol{\mu} + \left[\frac{\mathbf{x}^{\mathrm{T}}\mathbf{m}}{\mathbf{x}^{\mathrm{T}}\mathbf{V}\mathbf{x}} - \frac{\mathbf{x}^{\mathrm{T}}\mathbf{m}}{\mathbf{x}^{\mathrm{T}}\mathbf{V}\mathbf{x}} \cdot \frac{\mathbf{x}^{\mathrm{T}}\mathbf{\Sigma}\mathbf{x}}{\mathbf{x}^{\mathrm{T}}\left(\mathbf{V} + \mathbf{\Sigma}\right)\mathbf{x}} - \frac{\mathbf{x}^{\mathrm{T}}\boldsymbol{\mu}}{\mathbf{x}^{\mathrm{T}}\left(\mathbf{V} + \mathbf{\Sigma}\right)\mathbf{x}}\right] \cdot \left(\mathbf{\Sigma}\mathbf{x}\right)$$

$$= \boldsymbol{\mu} + \left(\frac{\mathbf{x}^{\mathrm{T}}\left(\mathbf{m} - \boldsymbol{\mu}\right)}{\mathbf{x}^{\mathrm{T}}\left(\mathbf{V} + \mathbf{\Sigma}\right)\mathbf{x}}\right) \cdot \left(\mathbf{\Sigma}\mathbf{x}\right).$$

Thus, the location parameter in (15) is a special case of this more general result when using $\mathbf{x} = \mathbf{1}$ and noticing again that $\mathbf{\Sigma}\mathbf{1} = \boldsymbol{\sigma}^2$ and $\mathbf{V}\mathbf{1} = \boldsymbol{\lambda}$, respectively. It can be seen that the Gaussian product updates the prior location in the direction $\mathbf{\Sigma}\mathbf{x}$. This is because the likelihood is translation invariant wrt. all directions perpendicular to $\mathbf{x}$ (i.e. the hyper plane determined by the normal vector $\mathbf{x}$).

The two posterior approximations $q_0$ and $q_{\mathrm{mix}}$ as well as the corresponding likelihood approximation is visualised in Fig. 3 for two different parametrisations (cf. Sec. 4.3).

### D.3 True posterior and optimal invariance-abiding parameters

For the linear model with a single observation $y$ and inputs $\mathbf{x}$, the true posterior $p(\mathbf{w} \mid \mathbf{x}, y)$ follows from the standard Bayesian update equation for Gaussian linear models (cf. (27) in App. B with $\mathbf{A} = \frac{1}{K} \cdot \mathbf{x}^{\mathrm{T}}$):

$$p(\mathbf{w} \mid y, \mathbf{x}) = \mathcal{N}\left(\mathbf{w}; \mathbf{m}^*_p, \mathbf{V}^*_p\right), \quad \mathbf{V}^*_p = \left(\frac{1}{K^2\sigma_y^2}\mathbf{x}\mathbf{x}^{\mathrm{T}} + \mathbf{\Sigma}^{-1}\right)^{-1}, \quad \mathbf{m}^*_p = \mathbf{V}^*_p\frac{y}{K\sigma_y^2}\mathbf{x}. \quad (74)$$

For $N$ observations $Y := \{y^{(n)}\}_{n=1}^N$ with identical input $\mathbf{x}$, the posterior is

$$p\left(\mathbf{w} \mid Y, \mathbf{x}\right) = \mathcal{N}\left(\mathbf{w}; \mathbf{m}^*_p, \mathbf{V}^*_p\right), \mathbf{V}^*_p = \left(\frac{N}{K^2\sigma_y^2}\mathbf{x}\mathbf{x}^{\mathrm{T}} + \mathbf{\Sigma}^{-1}\right)^{-1}, \mathbf{m}^*_p = \mathbf{V}^*_p\frac{\sum_{n=1}^N y^{(n)}}{K\sigma_y^2}\mathbf{x}. \quad (75)$$

We want to relate the true posterior (75) to the parameters of the invariance-abiding posterior $q_{\text{mix}}(\mathbf{w}; \boldsymbol{\theta})$. It suffices to consider the special case $\mathbf{x} = \mathbf{1}$ from the main text. We will use the form

$$q_{\text{mix}}(\mathbf{w}) = \mathcal{N}\left(\mathbf{w}; \left(\boldsymbol{\Sigma}^{-1} + \mathbf{V}_{\text{mix}}^{-1}\right)^{-1}\left(\boldsymbol{\Sigma}^{-1}\boldsymbol{\mu} + \mathbf{V}_{\text{mix}}^{-1}\mathbf{m}_{\text{mix}}\right), \boldsymbol{\Sigma}^{-1} + \mathbf{V}_{\text{mix}}^{-1}\right) \qquad (76)$$

For the precision matrix $\boldsymbol{\Sigma}^{-1} + \mathbf{V}_{\text{mix}}^{-1}$, using the form in (72) with $\mathbf{x} = \mathbf{1}$ gives

$$\mathbf{V}_{\text{mix}}^{-1} = \frac{1}{\mathbf{x}^{\mathsf{T}}\mathbf{V}\mathbf{x}}\mathbf{x}\mathbf{x}^{\mathsf{T}} = \frac{1}{\mathbf{1}^{\mathsf{T}}\boldsymbol{\lambda}}\mathbf{1}\mathbf{1}^{\mathsf{T}}, \qquad (77)$$

since $\mathbf{V}\mathbf{1} = \boldsymbol{\lambda}$. Comparing this to (75), we see that $\boldsymbol{\Sigma}^{-1} + \mathbf{V}_{\text{mix}}^{-1}$ has the same structure as the precision matrix of the true posterior. By setting the optimal variances as $\mathbf{1}^{\mathsf{T}}\boldsymbol{\lambda}^* = \frac{K^2\sigma_y^2}{N}$, and using $\frac{K^2\sigma_y^2}{N} = \frac{K\sigma_y^2}{N} \cdot \mathbf{1}^{\mathsf{T}}\mathbf{1}$, we see that one possible choice for the optimal variance parameters is

$$\boldsymbol{\lambda}^* = \frac{K\sigma_y^2}{N} \cdot \mathbf{1}. \qquad (78)$$

We note that other vectors that are not proportional to $\mathbf{1}$ and also sum to $\frac{K^2\sigma_y^2}{N}$ are also valid optima.

Similarly, with $\boldsymbol{\mu} = \mathbf{0}$, and by noting that $\boldsymbol{\Sigma}^{-1} + \mathbf{V}_{\text{mix}}^{-1} = \mathbf{V}_p^{-1}$, we see that

$$\mathbf{m}^* = \frac{1}{N}\sum_{n=1}^{N} y^{(n)} \cdot \mathbf{1}. \qquad (79)$$

# E  Permutation invariance in Bayesian neural networks

Here we analyse the *permutation* invariance in BNNs. This invariance is independent of the data, persisting for any dataset size.

We first describe the set of permutation matrices corresponding to the transformations that leave the likelihood invariant, i.e. $t(\mathbf{w}, \mathbf{r}) = \mathbf{P_r}\mathbf{w}$. We ignore the biases for simplicity and describe these permutations first on a node/neuron level, then layer-wise for the weight matrices, and finally on the weight vector that is obtained by stacking all weight matrices. We will denote layers with indices $l$ and the number of layers by $L$. Layer-wise permutation matrices are then denoted as $\tilde{\mathbf{P}}_l$ and the corresponding weight matrices are denoted as $\mathbf{W}_l$. The permutation matrix that results when stacking all weights matrices into a vector $\mathbf{w}$ is denoted as $\mathbf{P}$. When necessary, one particular permutation matrix from the set of all possible permutation matrices for a given architecture is indexed with superscript $^{(i)}$, i.e. $\mathcal{P} = \{\mathbf{P}^{(i)}\}_i$ and $|\mathcal{P}| = \prod_{l=1}^{L-1} k_l!$, where $k_l$ is the number of nodes in layer $l$.

**Node permutations.**  Each hidden layer $\mathbf{z}_l \in \mathbf{z}_1, \ldots, \mathbf{z}_{L-1}$ is a set of nodes that can be permuted in $k_l!$ possible ways, relabelling the corresponding parameters attached to these nodes. Each of these $k_l!$ permutations per layer can be combined with any of the permutations of another layer. Each permutation matrix $\tilde{\mathbf{P}}_l$ for a layer $l$ corresponds to one of the unique orderings of nodes $\mathbf{z}_l$. For instance, the permutation matrix that reorders the first 3 nodes as $(z_{l,3}, z_{l,1}, z_{l,2})$ is

$$\tilde{\mathbf{P}}_l = \begin{pmatrix} 0 & 0 & 1 & \ldots \\ 1 & 0 & 0 & \ldots \\ 0 & 1 & 0 & \ldots \\ \ldots & \ldots & \ldots & \ldots \end{pmatrix}. \qquad (80)$$

The remaining entries in the matrix are ones on the diagonal and zeros elsewhere.

**Layer-wise permutation of weight matrices.**  Next, we describe how node permutations correspond to permutations of weight *matrices* in terms of permutations to the in- and outgoing weights. For one particular instance of permutations (omitting superscript $^{(i)}$) to each of the hidden layers, the corresponding weight matrices can be permuted as follows:

$$\forall l \in 1, \ldots, L: \quad \mathbf{w}_l' = \tilde{\mathbf{P}}_l \mathbf{W}_l \tilde{\mathbf{P}}_{l-1}^{\mathsf{T}}, \quad \tilde{\mathbf{P}}_0 = \tilde{\mathbf{P}}_L = \mathbf{I}. \qquad (81)$$

The identities corresponding to the first and last layers is because only hidden layer nodes can be permuted but not the data itself. Note also that each permutation matrix is applied to two weight matrices, since permutations to nodes of a particular layer correspond to the simultaneous permutation of the weight matrices from the preceding and subsequent layer.

**Permutation of stacked weight vectors.** The layer-wise formulation can be written in terms of the stacked weight vector $\mathbf{w} = [\text{vec}(\mathbf{W}_1), \ldots, \text{vec}(\mathbf{W}_{L+1})]^\mathsf{T}$, using $\text{vec}(ABC) = \left(C^\mathsf{T} \otimes A\right) \text{vec}(B)$:

$$\text{vec}(\mathbf{W}_l') = \left(\tilde{\mathbf{P}}_{l-1} \otimes \tilde{\mathbf{P}}_l\right) \text{vec}(\mathbf{W}_l) =: \mathbf{P}_{l,l-1} \text{vec}(\mathbf{W}_l), \tag{82}$$

where we denote the new permutation matrix that is applied to the vectorised weights with a bar and the subscripts correspond to the two successive layers. The overall permutation matrix corresponding to the entire weight vector $\mathbf{w}$ is given by forming the block-diagonal matrix

$$\mathbf{P} = \begin{pmatrix} \mathbf{P}_{1,0} & \mathbf{0} & \mathbf{0} & \ldots \\ \mathbf{0} & \mathbf{P}_{2,1} & \mathbf{0} & \ldots \\ \mathbf{0} & \mathbf{0} & \mathbf{P}_{3,2} & \ldots \\ \ldots & \ldots & \ldots & \ldots \end{pmatrix}, \tag{83}$$

where each denotes block-matrices of zeros with the respective dimensions and the remaining parts of the matrix are the identity on the diagonal and zeros elsewhere.

**Invariance gap.** Note again that the permutation invariance for BNNs is independent of the data. The invariance gap $\text{KL}\left[q_0 \,||\, q_{\text{mix}}\right]$ takes values in the range $[0, \ln|\mathcal{P}|]$, where $|\mathcal{P}|$ is the factorial number of modes. The gap takes the maximal value when each mode is completely separated from the other modes, and the gap is zero when both $q_0(\mathbf{w})$ and $q_{\text{mix}}(\mathbf{w})$ are identical. This is the case e.g. if $q_0(\mathbf{w})$ reverts to the prior $p(\mathbf{w})$ since all modes are then identical, i.e. $q_0(\mathbf{Pw}) = q(\mathbf{w})$.

# F Data-related bound on the mean-field KL divergence

Assume the regression setting described in Sec. 2.2, where we assume a finite dataset $\mathcal{D} = \{(\mathbf{x}^{(n)}, y^{(n)})\}_{n=1}^N$ and a regression setting with fixed homogeneous noise variance $\sigma_y^2$. We can further assume that the prior is chosen such that it induces a reasonably bounded output variance of the neural network, $\sigma_L^2\left(\mathbf{x}^{(n)}\right) := \mathbb{V}_{\mathbf{w} \sim p}\left[f(\mathbf{x}^{(n)}; \mathbf{w})\right]$, for some fixed input $\mathbf{x}^{(n)}$. We will then have a finite expected log-likelihood and the ELBO for a mean-field variational approximation $q_{\boldsymbol{\theta}}$ is

$$\mathcal{L}_{\text{ELBO}}\left(q_{\boldsymbol{\theta}}, \mathcal{D}\right) = \underbrace{\sum_{n=1}^N \mathbb{E}_{\mathbf{w} \sim q_{\boldsymbol{\theta}}}\left[\ln p(y^{(n)} \,|\, \mathbf{x}^{(n)}, w)\right]}_{\text{ELL}(q_{\boldsymbol{\theta}}, \mathcal{D})} - \text{KL}\left[q_{\boldsymbol{\theta}} \,||\, p\right]. \tag{84}$$

Let us now consider a hypothetical *worst-case* fit in terms of the ELBO objective. This is when the data is completely ignored with $q_{\boldsymbol{\theta}}(\mathbf{w}) = p(\mathbf{w})$, as any worse fit could be trivially improved by setting the approximation to the prior. This gives then the inequality (using (84))

$$\forall q_{\boldsymbol{\theta}}: \quad \mathcal{L}_{\text{ELBO}}\left(q_{\boldsymbol{\theta}}, \mathcal{D}\right) \geq \mathcal{L}_{\text{ELBO}}\left(p, \mathcal{D}\right) \;\Leftrightarrow\; \text{ELL}\left(q_{\boldsymbol{\theta}}, \mathcal{D}\right) - \text{KL}\left[q_{\boldsymbol{\theta}} \,||\, p\right] \geq \text{ELL}\left(p, \mathcal{D}\right). \tag{85}$$

Although we can not compute the expected log-likelihood for an arbitrary fit $q_{\boldsymbol{\theta}}$, we can quantify the upper bound given above, by considering the *best-case* fit $q^*$, i.e. a hypothetical optimum where the data is perfectly predicted up to the known observation noise variance $\sigma_y^2$. The resulting bound is then

$$\text{KL}\left[q_{\boldsymbol{\theta}} \,||\, p\right] \leq \text{ELL}\left(q^*, \mathcal{D}\right) - \text{ELL}\left(p, \mathcal{D}\right). \tag{86}$$

Next, we compute the two expected log-likelihood terms in (86). We first consider the worst-case, where we have

$$\text{ELL}(p, \mathcal{D}) = \sum_{n=1}^N \mathbb{E}_{\mathbf{w} \sim p}\left[\ln p(y^{(n)} | \mathbf{x}^{(n)}, \mathbf{w})\right] \tag{87}$$

$$= \sum_{n=1}^N -\frac{1}{2\sigma_y^2} \mathbb{E}_{\mathbf{w} \sim p}\left[\left(y^{(n)} - z_L\right)^2\right] - \frac{N}{2} \ln\left[2\pi\sigma_y^2\right], \tag{88}$$

where $\mathbf{z}_L$ is the noisy output of the neural network given by

$$z_L = f(\mathbf{x}^{(n)}; \mathbf{w}) + \epsilon, \quad \mathbf{w} \sim p(\mathbf{w}), \quad \epsilon \sim \mathcal{N}\left(0, \sigma_y^2\right). \tag{89}$$

Splitting the expectation of the quadratic term into variance and square of the expectation, we have

$$\mathbb{E}_{\mathbf{w}\sim p}\left[\left(y^{(n)}-z_L\right)^2\right]=\mathbb{E}_{\mathbf{w}\sim p}\left[y^{(n)}-z_L\right]^2+\mathbb{V}_{\mathbf{w}\sim p}\left[y^{(n)}-z_L\right]\tag{90}$$

Since the last layer computes $f(\mathbf{x}^{(n)};\mathbf{w})=\mathbf{w}_{L,1}^{\mathsf{T}}\mathbf{z}_{L-1}$ in the regression setting and the prior weights are independent of $\mathbf{z}_{L-1}$ and centred at zero, i.e. $\mathbb{E}_{\mathbf{w}\sim p}[\mathbf{w}_{L,1}]=\mathbf{0}$, it follows that

$$\mathbb{E}_{\mathbf{w}\sim p}\left[y^{(n)}-z_L\right]=y^{(n)}.\tag{91}$$

$$\mathbb{V}_{\mathbf{w}\sim p}\left[y^{(n)}-z_L\right]=\mathbb{E}_{\mathbf{w}\sim p}\left[z_L^2\right]=\sigma_L^2\left(\mathbf{x}^{(n)}\right)+\sigma_y^2,\tag{92}$$

where we denote the variance of the neural network outputs by $\sigma_L^2\left(\mathbf{x}^{(n)}\right):=\mathbb{V}_{\mathbf{w}\sim p}\left[f(\mathbf{x}^{(n)};\mathbf{w})\right]$.

Hence, the expected log likelihood under the prior is

$$\mathrm{ELL}(p,\mathcal{D})=\sum_{n=1}^N-\frac{1}{2\sigma_y^2}\left[\sigma_L^2\left(\mathbf{x}^{(n)}\right)+\sigma_y^2+\left(y^{(n)}\right)^2\right]-\frac{N}{2}\ln\left[2\pi\sigma_y^2\right]\tag{93}$$

$$=-\frac{1}{2}\sum_{n=1}^N\frac{\sigma_L^2\left(\mathbf{x}^{(n)}\right)+\left(y^{(n)}\right)^2}{\sigma_y^2}-\frac{N}{2}\left(1+\ln\left[2\pi\sigma_y^2\right]\right).\tag{94}$$

For the best-case fit in terms of the ELBO, we assume that we completely overfit and perfectly predict the data by putting the mean of the Gaussian exactly on the data, i.e.

$$p(y^{(n)}|\mathbf{x}^{(n)},\mathbf{w})=\mathcal{N}(y;y^{(n)},\sigma_y^2).$$

Then, we have

$$\mathbb{E}_{\mathbf{w}\sim q^*}\left[y^{(n)}-z_L\right]=0,\tag{95}$$

$$\mathbb{V}_{\mathbf{w}\sim q^*}\left[y^{(n)}-z_L\right]=\sigma_y^2.\tag{96}$$

Hence, the expected log likelihood of the best case fit is

$$\mathrm{ELL}(q^*,\mathcal{D})=\sum_{n=1}^N\mathbb{E}_{\mathbf{w}\sim q^*}\left[\ln p(y^{(n)}\,|\,\mathbf{x}^{(n)},\mathbf{w})\right]\tag{97}$$

$$=\sum_{n=1}^N-\frac{1}{2\sigma_y^2}\left[\sigma_y^2\right]-\frac{N}{2}\ln\left[2\pi\sigma_y^2\right]\tag{98}$$

$$=-\frac{N}{2}\left(1+\ln\left[2\pi\sigma_y^2\right]\right).\tag{99}$$

Taking the difference between (93) and (97), we obtain the result in (3),

$$\mathrm{KL}\left[q_{\boldsymbol{\theta}}\,\|\,p\right]\le\sum_{n=1}^N\frac{\sigma_L^2\left(\mathbf{x}^{(n)}\right)+\left(y^{(n)}\right)^2}{2\sigma_y^2}.$$