# OpenReview forum: "On the detrimental effect of invariances in the likelihood for variational inference"
_NeurIPS.cc/2022/Conference — NeurIPS 2022 Accept_

### Official Review · Reviewer_aga4 · 2022-07-10

**Rating:** 5
**Confidence:** 4
**Soundness:** 3 good
**Presentation:** 2 fair
**Contribution:** 2 fair

**Summary:**

This paper looks at why mean-field variational inference can collapse to the prior with little data / large models, motivated by observations of this effect with Bayesian neural networks in the field. They define an "invariance-abiding" posterior approximation that is a function of the mean-field approximation, and introduce conditions on its form so that the likelihood term is the same, and the KL-to-prior term can be analysed. They look at translation invariance in a Bayesian linear regression model, showing both mathematically and empirically that the mean-field version reverts to the prior as the number of parameters increases, while the invariance-abiding alternative does not. There is then an initial attempt to use similar analysis for neural networks.

**Questions:**

I have provided detailed questions and suggestions in the Strengths and Weaknesses section above.

---------------
After rebuttal:

I thank the authors for improving the writing and clarity. I have increased my score. I did not increase by more because I still have some missing intuition between the linear model and neural networks (again, I stress this is important because of how the paper currently sells this as why it is important). I do appreciate the attempts the authors made already. I hope the authors can close this gap in future work to validate the importance of their analysis in this work.

**Limitations:**

No potential negative societal impacts that I can see.

**Strengths And Weaknesses:**

Overall, I think this kind of analysis is extremely relevant and of significance to the Bayesian deep learning community. Although there has been some previous work in this area (eg Kurle et al and Coker et al), a lot more can still be done, and this is a big strength of this paper for me.

Although the maths and approach for invariance-abiding posterior approximations for the linear model were overall nice, it is a toy example. In my mind, the key insight would be relating this maths to a less toy model such as a deep linear network or a full neural network with non-linearities. This key link is missing, as Section 4.3 does not provide any takeaways or insights. I would strongly consider raising my score if the authors could provide any intuitions or takeaways in Section 4.3. For example, can we say how the KL gap grows (intuitively)? Or are there any simplifications one can make to provide any such statement? Or anything else. Note that I am looking for any small but relevant statement for this paper (and any deeper analysis can be future work). I want something so that I know that this analysis may be useful in the deep learning case; right now it is still possible that this analysis is not useful at all because everything is just completely intractable and does not transfer over to neural networks.

I would also consider changing my view if I am convinced that the analysis for the linear model is a strong contribution in itself, however, I currently do not view it as enough for a paper (and this is not how the paper is written either).

I thought the Abstract and Introduction were written very well. However, I found the maths in the rest of the paper difficult to follow, and had to read it a few times before understanding most of it. It was very easy to confuse what was 'mix' and not, and although there was an attempt to provide some intuition before writing some equations, I think this can be looked over again and made simpler to parse. I think spending time on this would make the paper much more accessible and hence impactful to the community. It did not help that some things were not clearly defined / there were some typos: (i) Line 71, 'y' undefined, (ii) missing 'ln' in front of p(w|D) in line 80, (iii) line 90 should be $\sigma_y^2$, (iv) missing close bracket in Equation after line 117, (v) what is 'f' in Equation 7?, (vi) presumably 'lambda' is the variance of g(w) -- this is in line 123 but the way it is written could also be that {m, lambda} are means and variances of the approximate q, and not of g, (vii) Line 191 Equation does not define beta, (viii) Equation 17, mu is undefined (presumably it is prior mean, so zero?), (ix) Equation 21 should be $m^*_{mix}$ not $q_{mix}$.

Finally, the authors may also be interested in the paper, "On the expressiveness of approximate inference in Bayesian neural networks", Foong et al, 2021.

(Note that I did not check the derivations/proofs in the supplementary material.)

---

> ### Author Response · Authors · 2022-08-02
> **Response to reviewer feedback**
>
> We thank the reviewer for his constructive feedback and suggestions.
>
> 1. “... In my mind, the key insight would be relating this maths to a less toy model such as a deep linear network or a full neural network with non-linearities. This key link is missing, as Section 4.3 does not provide any takeaways or insights…”
> Thank you for this suggestion. We agree that ideally this crucial step that should be sketched in this work, while future work can flesh out details. To this end, we provided a discussion for a potential inference algorithm that takes into account the translation invariance.
> Regarding, “For example, can we say how the KL gap grows (intuitively)?”, we realised that this part still comes short and we will expand on this question in the next days.
>
> 2. “I thought the Abstract and Introduction were written very well. However, I found the maths in the rest of the paper difficult to follow, and had to read it a few times before understanding most of it…“
> We re-wrote most sections of the paper, removing unnecessary details from the main text, and adding discussions of the results. We think that this already greatly improved the presentation and we appreciate further feedback.
>
> 3. “ It did not help that some things were not clearly defined / there were some typos…”
> Thanks a lot for pointing out these typos and missing definitions, we have corrected these.
>
> 4. “Finally, the authors may also be interested in the paper, "On the expressiveness of approximate inference in Bayesian neural networks", Foong et al, 2021.”
> This paper is indeed relevant, we now mention it in the related work.

---

> ### Author Response · Authors · 2022-08-09
> **Response to reviewer regarding link to BNNs**
>
> Thank you again for your feedback. We have used the feedback of all reviewers to greatly update the presentation of the paper.
> Your question "can we say how the KL gap grows (intuitively)?" is of course very relevant. Unfortunately, we can not answer this question for *translation invariance* yet. We acknowledge this as a limitation of the work as it currently stands. In future work, we plan to develop approximations or bounds that allow us to approximate the impact as well as correct for this invariance-gap (as we can do for the linear model).
> We can however say how the *invariance-gap* grows for node-permutation invariance, which we characterise in App. E of the the supplementary material: The invariance gap is between zero and the logarithm of the number of (equivalent) discrete modes induced by the permutation invariance. The number of modes is factorial in the number of neurons for each layer and multiplicative in the number of layers.
> The gap takes values on the order of hundreds of nats for modest architecture sizes.
> In preliminary work, we realised, however, that the rest of the KL regularisation term can still be e.g. an order of magnitude larger. This observation actually led to this work, where we investigated BNNs for further invariances (translation) and came up with the canonical model that implements (only) translation invariance and where we can positively answer the question whether this invariance has a detrimental effect.
>
> Intuitively however, by going from a linear model without hidden layers to a linear Bayesian Neural Network, we increase over-parametrisation/invariances, and we would expect that the invariance-gap can then only grow additionally.

---

### Official Review · Reviewer_11Q5 · 2022-07-11

**Rating:** 7
**Confidence:** 3
**Soundness:** 4 excellent
**Presentation:** 3 good
**Contribution:** 3 good

**Summary:**

The paper provides a novel take to the long-standing problem of posterior collapse in training mean-field variational Bayesian neural networks (VBNNs). It is argued that the invariances inherent to the architures of the network lead to posterior distributions with complicated structures that are hard to capture with mean-field variational families. A quantitative measure, referred to as the "invariance gap" is proposed to study how this invariance hinders variational inference. The translational invariances in Bayesian linear regression and deep neural network models are studied under this framework.

**Questions:**

- It is not very clear to me how Figure 2 should be interpreted: basically, we have two sets of optimal parameters: $\theta_0^*$ and $\theta_{\text{mix}}^*$. As dimensionality increase, the invariance gap $\text{KL}[q_0 || q_{\text{mix}}]$ dominates when you plug in $\theta_{\text{mix}}^*$ but not for $\theta_{0}^*$. How does one arrive at the conclusion that invariance-gap incentivises posterior collapse? What does it mean for the gap for the mean-field approximation to vanish?

- Does the proposed invariance-abiding variational family lead to less posterior collapse in practice? In the case where some but not all of the invariances are addressed with this mixture variational family, can we still see improvements nontheless?

- We know from the KL gap that collapsing to the prior can be a local minimum that is easy to obtain. However, are there any insights on the training dynamics that lead to this local minimum, instead of learning one posterior mode that "breaks the symmetry" of the system? From experience, even for problems without this type of invariance, bad hyperparameters and/or initialization could still lead to posterior collapse.

**Limitations:**

The biggest limitation of this work is the lack of practical procedures to address the invariance problem in practice, which the authors addressed clearly in the conclusion section.

**Strengths And Weaknesses:**

The paper presents a novel idea in a clear and mathematically precise manner. In-depth theoretical analysis is provided which supports the ideas presented. The background knowledge (Bayesian neural network and variational inference) is presented in a way that is easy to follow. I also like how the idea of the detriments of invariances are introduced and discussed point by point, gradually giving the reader a clearer picture of what is going on.

However, there are a couple of weak points later on in the paper. The theoretical results sections (subsections 4.1-4.3) are a bit unnecessarily dense with technical details, with not enough exposition and discuss on the significance of these results. Subsection 4.3 in particular has a lot of formulae but very little discussion of the significance of these results. I wish that the theoretical results can be presented more concisely, giving more room for more insightful discussions, such as potential solutions to the invariance problem.

Overall the paper is well written and the contributions seem solid, with some room for improvement in the theoretical sections.

---

> ### Author Response · Authors · 2022-08-02
> **Response to reviewer feedback**
>
> Thank you for your constructive feedback. We gladly address your questions below:
>
> 1. “The theoretical results sections (subsections 4.1-4.3) are a bit unnecessarily dense with technical details, with not enough exposition and discuss on the significance of these results…”
> We appreciate this feedback and agree with you. We removed some details that were not necessary and added several short discussions.
>
> 2. “It is not very clear to me how Figure 2 should be interpreted...”
> Thank you for pointing out inaccurate statements. We updated the figure caption. What we can see is that the optimal parameters for the mean-field approximation (optimised through the respective ELBO) can not coincide with the optimal parameters of the invariance-abdiding approximation (also optimised through its respective ELBO). And in Eq. (19) we can also see that the optimal variances grow quadratically with K, whereas the prior variances is chosen to grow linearly with K (so as to induce the same predictive distribution independent of K). This means that as K tends to infinity, the influence of the likelihood vanishes compared to the prior.
>
> 3. “Does the proposed invariance-abiding variational family lead to less posterior collapse in practice? In the case where some but not all of the invariances are addressed with this mixture variational family, can we still see improvements nontheless?”
> This is an interesting question that is difficult to answer. For our linear model, since the invariance-abiding approximation can perfectly approximate the true posterior it does prevent posterior collapse. For non-linear models we can not yet answer this question. First, note that the invariance-gap is not generally tractable for non-linear models but needs to be approximated. In preliminary work (not published), we approximated the invariance-gap that results from the node-permutation invariance (described in App. F). We found that approximating the gap for only this invariance was not sufficient. We then realised that BNNs also exhibit translation invariance that needs to be addressed. There could be even further invariances in BNNs though. As the first step in this research direction, we focussed on the linear model case that we designed specifically such that it exhibits only translation invariance, allowing us to evaluate its detrimental effect without worrying about additional invariances that we might have overlooked.
>
> 4. “We know from the KL gap that collapsing to the prior can be a local minimum that is easy to obtain. However, are there any insights on the training dynamics that lead to this local minimum, instead of learning one posterior mode that "breaks the symmetry" of the system?”
> In the linear model case, the optima are actually computed exactly; there are no problems with local optima. We show e.g. that in the optimal mean-field approximation, the likelihood has variance parameters that grow quadratically with the number of parameters K, while the prior grows only linearly. The optimal mean-field posterior resulting from the product of prior and likelihood will therefore revert to the prior. In our experience with BNNs, we have also observed that BNNs initialised at the MAP estimates escape from these minima (reverting partially to the prior), thereby obtaining better values for the ELBO, but significantly worse predictions. What is the model that you refer to which does not exhibit this type of invariance?

---

### Official Review · Reviewer_ZF6G · 2022-07-11

**Rating:** 6
**Confidence:** 3
**Soundness:** 3 good
**Presentation:** 2 fair
**Contribution:** 3 good

**Summary:**

This paper posits that the main reason that overparameterized Bayesian neural networks tend to collapse to the prior distribution is attributed to the fact that there exist invariances in the likelihood with respect to model parameters. Conversely, the paper proposes that should these invariances be accounted for and handled properly then this behavior can be avoided. The paper develops an analysis framework to tackle this as well as investigates a case-study of accommodating translational invariance in a particular linear model and a general feed-forward Bayesian neural network.

**Questions:**

1. On line 92, what does the $k+1$ in $\sigma_{k+1}$ indicate?
2. On lines 199 and 200, you assume that "all variances and means take the same value for the approximation and the prior". Can you expand on why this is okay to make this assumption?
3. To better my understanding, what exactly is holding back a practitioner from trying to find $\theta^*_\text{mix}$ (Eq. 19a) for say a generic Bayesian neural network? In other words, what is the immediate roadblock to optimizing for this objective that requires further research?
4. When deriving the invariance gap in the Appendix, on Eq. 48 after line 470 Jensen's inequality is applied to move a logarithm outside the inner integral; however, the inequality is used as an equality. I have two concerns with this. The first being that the logarithm essentially "skipped past" a scalar constant (i.e., what happened: $\int c \log(x) dx \rightarrow \log \int cx dx$), which I do not believe is applicable for Jensen's inequality. Second, Jensen's inequality is only an equality if the function being brought outside of the integral is affine (which $\log$ is not) or that the remainder of the integrand is constant (which $Z_\text{mix}g_\text{mix}(t^{-1}(\mathbf{w},f(\mathbf{r}))d\mathbf{w}$ is not w.r.t. $\mathbf{w}$). Can you please expand on why this was a valid operation to do as I may have missed something. Should this not have been valid, how does this then effect the invariance gap and subsequent findings?

**Limitations:**

The authors have adequately addressed the work's limitations.

**Strengths And Weaknesses:**

The paper proposes a novel hypothesis to the often cited phenomenon of a Bayesian neural network's posterior collapsing to the prior distribution. Backing this up a new bound is developed, the invariance gap, that serves to illustrate in what scenarios will a model collapse (resulting in a invariance gap of 0). I believe these findings to be significant and meaningful to the community. That being said, I found the work to be very dense and hard to follow at times. Particularly the paragraph titled "Likelihood and Posterior approximations" starting on line 108 was difficult and required rereading many times over to follow.

As for the work done on the translation invariant, the work largely seemed meaningful. While it would have been nice to see it actually applied in an experimental setting, I do recognize that this isn't in the intended scope of the paper.

---

> ### Author Response · Authors · 2022-08-02
> **Response to reviewer feedback**
>
> We thank the reviewer for the helpful feedback and are glad to answer open questions:
>
> 1. “I found the work to be very dense and hard to follow at times. Particularly the paragraph titled "Likelihood and Posterior approximations" starting on line 108 was difficult and required rereading many times over to follow.”
> Thank you for your feedback; we have made significant updates to the paper presentation. We hope that it is much more comprehensible now. We appreciate further feedback.
>
> 2. “On line 92, what does the k+1 in σk+1 indicate?”
> Sorry for the confusion, we introduced this typo while shortening the section. The index is now changed to “L” which denotes the last (output) layer of the neural network (which is now also mentioned in the text).
>
> 3. “On lines 199 and 200, you assume that "all variances and means take the same value for the approximation and the prior". Can you expand on why this is okay to make this assumption?”
> We added a concise discussion about this. There is no difference for q_mix, since it integrates over all valid translations, and the variances for the optimal g_0 are proportional to the vector 1 (in this simple setting) due to the prior having variance proportional to 1 as well.
>
> 4. “To better my understanding, what exactly is holding back a practitioner from trying to find θmix*(Eq. 19a) for say a generic Bayesian neural network? In other words, what is the immediate roadblock to optimizing for this objective that requires further research?”
> One problem is that computing q_mix or the ELBO objective is not generally tractable. As we have shown, the simpler q_0 can be sufficient (equivalent) for predictions, but we need to take into account the invariance gap in the objective. For non-linear models, we will have to find a tractable approximation.
> The second problem is that for Bayesian neural networks, it is not obvious which invariances they exhibit. One type of invariance in BNNs is the node permutation invariance described in the appendix. In preliminary work, we approximated the invariance-gap resulting from permutations only. However, we then realised that BNNs additionally exhibit translation invariance (which must be taken into account).
> Both of these problems provided the motivation for our analysis of the over-parametrised linear model, because we know exactly the invariance (by design), we can also model it exactly (without approximations). We could thus observe its detrimental effect in isolation from approximation errors and knowing that we took into account all existing invariances.
> The BNN section in our paper then provides the basis for novel approximation methods that also take into account the translation invariance.
>
> 5. “When deriving the invariance gap in the Appendix, on Eq. 48 after line 470 Jensen's inequality is applied to move a logarithm outside the inner integral; however, the inequality is used as an equality. I have two concerns with this…”
> Thank you for pointing this out. We now provide a simpler derivation that does not need Jensen’s inequality.

---

> > ### Comment · Reviewer_ZF6G · 2022-08-09
> > **Response**
> >
> > Thank you for the in depth replies to my comments and questions. I believe with the changes promised / made that the paper is a much better piece of work than before. I have updated my score to reflect this (5 $\rightarrow$ 6).

---

### Official Review · Reviewer_BBQR · 2022-07-18

**Rating:** 4
**Confidence:** 2
**Soundness:** 2 fair
**Presentation:** 1 poor
**Contribution:** 3 good

**Summary:**

This paper evaluates the impact of the influence of the invariances in the likelihood function on the ELBO, when performing Variational Bayes. The authors propose to enforce posteriors to be invariant to a few transformations (permutations of parameters, translations). Compared to the usual posteriors, the posteriors obtained this way achieve (of course) the same performance in terms of likelihood, but lead to a much larger KL term in the ELBO bound. That is, changing a posterior $q\_0$ to an invariant posterior $q\_{\mathrm{mix}}$ increases the distance of the posterior to the prior $p$.

This observation can be useful to explain and fix the "posterior-to-prior collapse" effect (see [4], "Wide Mean-Field Variational Bayesian Neural Networks Ignore the Data", Coker et al., 2021). According to [4], when performing Variational Bayes (VB) on a Neural Network (NN) whose number of neurons tends to infinity, the variational posterior (i.e., the distribution optimizing the ELBO) tends to the prior. The authors of the present work claim that their study will be useful in the future to understand and overcome this effect, since they are able to affect the KL term of the ELBO without changing its likelihood term.

**Questions:**

-

**Limitations:**

-

**Strengths And Weaknesses:**

## Strengths

The impact of the invariances of a NN on the optimization process and the best achievable validation loss is a very hot topic, with a wide range of works spanning many aspects of the NNs, for instance: implicit bias of the SGD [A], regularization effect of overparameterization [B].

Moreover, this work is in line with [4], which sheds light on inconsistencies in VB applied to NNs.

## Weaknesses

The major weakness of the paper is its readability.

### Readability

This paper is written as a succession of statements and sketches of proofs, without highlighting anything. Providing technical details in the main part of the paper is a good idea. But, at least, the authors should be clearer about the nature of each of their paragraphs.

For instance, definitions may be put into "definition" environments with "Definition X" written in bold. Same with claims, which may be put into "Theorem" or "Proposition" environments, and sketches of proofs, which may be put into "Proof" environments.

The term "mean-field" is not well defined by the authors. For me, "mean-field variational Bayes" means "VB where the posterior distribution is a product of distributions over each variable". That is, independence between variables in the posterior. I did not manage to check whether I was right or not by reading the paper.

Besides, I was not able to fully understand the paragraph lines 108-128. In particular, the function $g\_0(w, \theta)$ "approximating" $l(w, \mathcal{D})$ looks like a way to introduce the "variationalized" model, but there is not enough information and precise definition to be sure of that. Besides, there is no definition of $l(w, \mathcal{D})$.

Several mathematical objects are not well or explicitly defined, which decreases the readability of the paper. For instance:
 * Equ. (7): $Z\_0(\mathbf{r})$ is not defined, even in the appendix;
 * Equ. (7): what is $f$?

Notes:
 * denoting a variance by "$\sigma$": "$\sigma^2$" must be used instead;
 * headers of LaTeX "paragraphs" should not end with ":", but with "." or nothing;
 * what is the signification of the "!" over the "=" (see Equ. (7), (8) and many equations in the appendix)?
 * line 166: "models" -> "model";
 * line 116: "see 4" -> "see Section 4"

### Correctness

See appendix, Equations (46)-(50):
 * $\mathrm{d}\mathbf{w}$ and $\mathrm{d}\mathbf{r}$ must be swapped, to be consistent with the integration order;
 * the authors must use parentheses when using $\ln$ on several factors: prefer $\ln(ab)$ instead of $\ln a b$;
 * the use of Jensen's inequality-equality between (47) and (48) is very obscure and needs an explanation; if $g\_{\mathrm{mix}}(\cdots)$ is constant in $\mathbf{r}$, there is no need to use Jensen (and we recover the authors' result), if it is not constant in $\mathbf{r}$, how do the authors justify the "Jensen's equality" they used?

About the "infinite variance Gaussian distribution" (see lines 115-116 and Equation line 191): there is no theoretic justification or discussion about this choice. The main question to answer is: would we get the same result with any sequence of distributions tending to an improper "uniform distribution" over $\mathbb{R}$?

## References

[A] "In Search of the Real Inductive Bias: On the Role of Implicit Regularization in Deep Learning", Neyshabur et al., 2015

[B] "Stochastic complexity of Bayesian networks", Yamazaki and Watanabe, 2002

---

> ### Author Response · Authors · 2022-08-02
> **Response to reviewer feedback**
>
> Thank you for the detailed review, these were very helpful suggestions. We addressed the problems that were raised:
>
> 1. “This paper is written as a succession of statements and sketches of proofs, without highlighting anything.”
> Thank you, we agree with your feedback. We therefore re-wrote many sections, removing unnecessary details and adding several discussions of the results.
>
> 2. “definitions may be put into "definition" environments …”.
> Thank you for this suggestion. We have now formulated the posterior predictive equivalence and the invariance-gap in lemma environments that we prove subsequently (see in Appendix D). We will also re-formulate the corresponding part of the main text in the next update and we will gladly implement further suggestions.
>
> 3. “The term "mean-field" is not well defined by the authors.”
> We use the same definition as you mentioned, though we would add that the product of distributions is typically Gaussian. We clarified this in the first paragraph of the introduction and provided a formal definition in the background section. We also avoid using additional terms such “factorising distribution” to avoid confusion.
>
> 4. “Besides, I was not able to fully understand the paragraph lines 108-128. In particular, the function g0(w,θ) ‘approximating’ l(w,D) looks like a way to introduce the ‘variationalized’ model”. We re-wrote these paragraphs to make this part much clearer. g_0 is indeed a (mean-field) variational approximation of the likelihood. We expanded on the intuition that the likelihood might itself be composed of simpler functions and that approximating one single mode may be sufficient. For instance, the node permutation invariance in BNNs induces a factorial number of equivalent discontinuous modes (discussed in Appendix E), and the likelihood of the Bayesian linear regression model that exhibits translation invariance can also be (perfectly) constructed from a Gaussian mean-field likelihood.
>
> 5. “the use of Jensen's inequality-equality between (47) and (48) is very obscure and needs an explanation” and “dw and dr must be swapped…”. Thank you for pointing out this inaccuracy. We have updated the derivation, avoiding Jensen’s inequality (as an equality) and using the correct integration order. We think that the derivation is easier now. We also added a concise explanation for the relevant steps in each line.
>
> 6. “About the "infinite variance Gaussian distribution" (see lines 115-116 and Equation line 191): there is no theoretic justification or discussion about this choice. The main question to answer is: would we get the same result with any sequence of distributions tending to an improper "uniform distribution" over R?”
> This is an interesting question that we unfortunately can not answer. We added a brief explanation that motivates our choice though: since g_0 is Gaussian, we can compute the marginalisation exactly. Other parametric distributions that have a different form would not allow us to use the same analytical results.
>
> 7. Thank you also for spotting many additional typos and missing definitions. We have addressed each of these, except that we currently still use \sigma to denote the variance rather than standard deviation. We will update the paper again and either use a new symbol or write \sigma^2 as suggested.
>
> Thank you again for your time, we will be glad to engage further if you have additional suggestions.

---

### Meta-Review · Area_Chair_9BVj · 2022-08-27

**Recommendation:** Accept
**Confidence:** Certain

**Metareview:**

This paper seeks to understand and explain why variational Bayes seems to underperform (if not fail completely) in overparameterized models such as Bayesian NNs.  The proposed explanation is an additional gap in the marginal likelihood bound that results from these invariances.  The paper demonstrates the gap in detail for a linear model with translation invariance.  The reviewers had a favorable opinion of the work, having only the criticisms of (1) clarity of writing and (2) questioning if the paper can say anything concrete about Bayesian NNs.  The authors have substantially revised the work, and I and the reviewers agree that the clarity now meets the bar for publication.  I think the second criticism is still valid, and the paper would be better served to be billed as a study of overparameterized models.  Yet, the paper's merits outweigh this downside (which I encourage the authors to improve upon before the camera ready).

Also, this paper is related enough to warrant inclusion in the related work, as it seeks to directly address the symmetries algorithmically:
Moore, David A. "Symmetrized variational inference." In NIPS Workshop on Advances in Approximate Bayesian Inference, vol. 4, p. 31. 2016.

**Award:**

No

---

### Decision · Program_Chairs · 2022-09-14

Accept